# Negative Parenting Style and Perceived Non-Physical Bullying at School: The Mediating Role of Negative Affect Experiences and Coping Styles

**DOI:** 10.3390/ijerph19106206

**Published:** 2022-05-19

**Authors:** Houyu Zhou, Qinfei Wang, Shuxu Yu, Quanquan Zheng

**Affiliations:** 1Department of Psychology, Jing Hengyi School of Education, Hangzhou Normal University, Hangzhou 311121, China; 2Department of Curriculum and Teaching, Jing Hengyi School of Education, Hangzhou Normal University, Hangzhou 311121, China; 2018211101177@stu.hznu.edu.cn (Q.W.); 2018211101185@stu.hznu.edu.cn (S.Y.); 3Department of Psychology and Behavioral Science, Zhejiang University, Hangzhou 310015, China; qqzheng@zju.edu.cn

**Keywords:** perceived school non-physical bullying, rejection parenting styles, negative affect experiences, negative coping styles, mediating role, China

## Abstract

At present, school bullying incidents frequently occur, attracting increased attention from researchers. In this study, we attempt to explore the impact of parenting styles on perceived school non-physical bullying. Four hundred ninety-two students in the fifth and sixth grades of eight primary schools in Zhejiang province were surveyed. To control any potential confounding factors, a randomized sampling survey method was used to distribute questionnaires. The results showed that negative affect experiences, negative coping styles, negative family parenting styles, and the perceived school non-physical bullying were all positively correlated with each other (*p* < 0.05). Perceived verbal bullying differed significantly by gender, grade, and only/non-only children (*p* < 0.05). Perceived relationship bullying significantly differed between grades (*p* < 0.05). The gender difference in perceived cyberbullying also reached a significant level (*p* < 0.05). The rejection parenting style was shown to be an important factor that may be associated with students’ perceived school non-physical bullying; it was observed to be directly associated with students’ perceived school non-physical bullying and indirectly associated with students’ perceived school non-physical bullying by influencing negative affect experiences and negative coping styles. In conclusion, negative affect experiences and coping styles may have a chain-like mediating effect between the rejection parenting style and students’ perceived school verbal bullying. Moreover, negative affect experiences may have a partial mediating effect between the rejection parenting style and students’ perceived school cyberbullying, relationship bullying, and non-physical bullying total scores. This study provides first-hand empirical data support for schools, families, and education authorities to guide and manage non-physical bullying incidents in schools. They also provide a theoretical basis for subsequent related research in the field of non-physical bullying.

## 1. Introduction

Olweus first proposed the term “bullying,” which he defined as a deliberate, continuous, and regular negative action taken by a single student or a group of students against a specific student or group of students to cause physical and psychological harm [1]. School bullying is unwelcome offensive behavior, and victims (The victims here include targets of bullying and the victim. “Victim” will be used throughout this article rather than “target of bullying” for brevity’s sake.”) may endure physical, psychological, social, or educational harm [2]. Research showed that children who were identified as the victims of school bullying face higher risks of psychological stress and adaptation problems during adolescence and adulthood [3,4,5,6].

Depending on whether physical bullying is used, the main school bullying behaviors can be divided into school physical bullying and school non-physical bullying. School non-physical bullying refers to forms of school bullying other than physical bullying, such as verbal bullying, relationship bullying, and cyberbullying [7]. Verbal bullying refers to direct verbal attacks, including nicknames, curses, humiliation, and sarcasm [8]. Relationship bullying is an act of deliberate damage to an individual’s self-esteem and social status, including threats to end a friendship, social exclusion, spreading rumors, and deliberately ignoring the victim [9]. Cyberbullying is a harmful activity carried out by students on the Internet through language, pictures, videos, and other forms, such as sending threatening, unpleasant, or unwelcome texts [10]. The roles of school non-physical bullying are the initiator of the non-physical bullying, the bully, and the recipient of the non-physical bullying, the bullied [11]. Perceived school non-physical bullying refers to a student or a group realizing and believing that they have been subjected to school non-physical bullying, which causes his or their mental health to suffer a certain degree of adverse effects. Research shows a link between perceived verbal bullying and self-harm: Perceived verbal bullying in non-physical bullying greatly increases the likelihood of self-harm in young people. [12]. Perceived cyberbullying also impacts the victim’s social and mental health [13] and it is associated with mental health problems, such as anxiety, depression, suicidal tendencies [14], alcohol addiction [15], and smoking addiction [16], and low quality of life. Perceived relationship bullying not only causes more emotional problems for the victim but also leads to some externalizing problems, increasing the likelihood of being the target of repeated bullying [17,18].

To investigate the current situation of non-physical bullying among urban primary school students in China, an interview study was conducted after the literature study (see the research methods section for research design and basic information on research objects). From the interview research, we found that when faced with the same bullying behaviors, such as "others call me bad nicknames, or make fun of and ridicule me,” different interviewees had different interpretations of these similar or identical behaviors. Some respondents thought they were hurt a lot. Some thought it was a funny joke, and some thought it was just normal human interaction. The interviewees’ interpretations and responses to non-physical bullying among classmates were quite different among the upper primary school students. Whether students perceived non-physical bullying depended not only on the behaviors themselves but also appeared to be influenced by other factors.

Based on the results of interviews and previous literature research, this study attempts to explore the reasons for these differences. By exploring the causes and clues of perceived non-physical bullying behaviors, we can improve various behavioral factors that are easy to induce such phenomena in family upbringing and school education and teaching to effectively prevent the causes of poor psychological development of students. The results of this study can enrich the research results of primary school students’ non-physical bullying from the perspective of family parenting style, provide empirical data support for the follow-up theoretical research in related fields, and also give specific operational ideas and basis for families and schools to intervene in non-physical bullying effectively. For achieving the above objectives, we explored the mental mechanism of the perceived school non-physical bullying from the following aspects: parenting styles, negative affect experiences, and negative coping styles.

### 1.1. Negative Parenting Styles

The family parenting style is an important factor that influences the occurrence of non-physical bullying at school [19,20,21]. A negative parenting style refers to a kind of stable negative behavior tendency that accompanies parents in the process of educating their children [22], including the three dimensions of rejection, overprotection, and anxiety. Evidence shows that when children experience positive parenting styles, parents may be more sensitive to their children’s needs, thereby improving their relationships with their children. Improvements in this relationship can cultivate children’s emotional regulation and problem-solving abilities [23,24], as well as reduce negative affect experiences and negative coping styles. Studies have found that the rejection parenting style is significantly positively correlated with children’s psychological disorders, conduct problems, hyperactivity, and attention deficits, but it is negatively correlated with adolescent prosocial behaviors significantly [25,26]. The essence of Carl Rogers’s theory is that acceptance and unconditional positive regard are the basis for mental health and that rejection is the basis for psychological disturbances [27]. Rejection may not only hurt the self-concept and undermine children’s feelings of relatedness to their parents but also result in the sense of alienation from the child’s authentic self. Children who experience the rejection parenting style are less likely to establish positive relationships with others outside the family, are more likely to be bullied [28,29], and thus are more likely to be perceived as the target of non-physical bullying at school [30]. Previous studies have also found that children who are overprotected by their parents may not develop qualities such as autonomy and advocacy; overprotection in the negative parenting style increases the probability that children perceive school bullying, including non-physical bullying [31]. The rejection and anxious negative parenting styles are related to children’s perceived cyberbullying in the context of school non-physical bullying. These excessive or neglected parenting styles comprise one of the variables regarding perceived cyberbullying [32].

Therefore, this study proposes:

**Hypothesis** **H1.**
*Negative parenting styles are positively related to perceived school non-physical bullying.*


### 1.2. Negative Affect Experiences

Negative affect experience is a dimension of subjective emotion, which is a kind of painful and unpleasant psychological experience [33] that includes fear, panic, disgust, guilt, and tension. A negative parenting style influences children’s negative affect experiences. Compared with the supportive, positive parenting style, the rejection negative parenting style lowers a child’s ability to deal with emotions in social situations [34]. A negative parenting style, which includes severe obedience and punishment, can hinder children’s ability to regulate their emotions. For example, a mother’s negative reaction (e.g., neglect or punishment) to a child’s anger can prevent the child from learning to resolve the negative affect experience of anger [35]. Parents’ negative reactions to their children’s emotions, such as pain, fear, tension, and sadness, are usually considered to be related to their children’s negative emotional results [36,37]. One possible reason is that children who show negative affect experience gradually learning to hide their emotions, but they feel anxious and nervous when the emotions are aroused because there is a repeated connection between the rejection and punishment of negative parenting styles and the ability to express emotions. This kind of negative affect experience leads to anxiety, and tension may manifest itself as intrinsic motivation [38]. It was also shown that after being raised with a negative parenting style, including being insulted and shouted at, children have negative emotional experiences, such as tension and feelings of inferiority.

These negative affect experiences make it difficult for children to effectively protect themselves from various attacks from school peers, which consequently leads to the emergence of non-physical bullying victims [39]; that is, family victimization in childhood increases the probability of the perceived school bullying.

Therefore, this study proposes:

**Hypothesis** **H2.**
*Negative parenting styles are positively related to negative affect experiences.*


**Hypothesis** **H3.**
*Negative parenting styles can affect perceived school non-physical bullying by influencing negative affect experiences.*


### 1.3. Negative Coping Styles

Coping styles are the cognitive and behavioral methods adopted by individuals in a stressful environment to alleviate the negative effects of stress [40]. They are important mediating factors in the process of dealing with psychological stress. Negative coping styles include practices such as accepting reality, imagining miracles, self-comforting, and trying to forget events. Due to a lack of experience in coping with social problems, students tend to adopt negative coping styles that are not conducive to coping with challenging problems [41]. This is consistent with the conclusion made by other scholars, which posit that positive coping styles and anxiety symptoms found by other scholars are negatively correlated [42]. Some researchers found that emotional problems are positively related to coping styles. Negative emotions can make individuals adopt coping styles such as avoidance and denial [43]. In addition, negative coping styles can affect students’ perceived non-physical bullying because victims often have poor self-esteem and think of themselves as losers who are unattractive, unintelligent, and insignificant. These negative perceptions cause the victim to wrongly attribute bullying to themselves [44]. Furthermore, coping strategies can make children attribute their discrimination and bullying to personal characteristics or identities [45]. For example, among students in grades 4 to 6, compared to peers who attributed the victimization to other personal characteristics or identities, young people who attributed victimization to their ethnic identity were found to be less likely to use support-seeking coping styles and more likely to use negative coping styles [46,47]. This means that victims who have undergone non-physical bullying are usually unwilling to reveal bullying events, and these negative coping styles of accepting reality, trying to forget, and not disclosing their victimization may attract bullies and lead to the repeated targeting of victims. Previous researchers pointed out that the most common coping styles adopted by students aged 13 to 16 to deal with bullying are negative and include ignoring the bully (73%) and walking away (70%). They also reported other coping styles, such as defending oneself (68%), which was especially used among boys, and fighting back (28%). More than a quarter of children (26%) reported that they had passively accepted their circumstances [48,49].

Therefore, this study proposes:

**Hypothesis** **H4.**
*Negative affect experiences are positively related to negative coping styles.*


**Hypothesis** **H5.**
*Negative coping styles have a mediating effect between negative parenting styles and students’ perceived school non-physical bullying.*


Based on the results of the above literature research, we found a chain of causality. There has been little research on the effects of parenting styles, negative affect experiences, and negative coping styles on the perceived school non-physical bullying. Therefore, we attempted to explore the associations of parenting style, negative affect experience, and negative coping styles on the perceived school non-physical bullying among senior primary school students, where we hypothesized that negative emotional experiences and negative coping styles have mediating effects on the influence of family parenting styles on the perceived school non-physical bullying. The study aims to draw corresponding conclusions and enrich empirical research in relevant fields by analyzing the questionnaire data regarding pupils’ negative coping styles, negative affect experiences, parenting styles, and non-physical bullying scores. This study can provide a reference for school management and family educational intervention of high-grade students’ perceived school non-physical bullying.

Therefore, this study proposes:

**Hypothesis** **H6.**
*Negative affect experiences and negative coping styles have a mediating effect between negative parenting styles and students’ perceived school verbal bullying.*


**Hypothesis** **H7.**
*Negative affect experiences and negative coping styles have a mediating effect between*
*negative*
*parenting styles and students’ perceived school relationship bullying.*


**Hypothesis** **H8.**
*Negative affect experiences and negative coping styles have a mediating effect between negative parenting styles and students’ perceived school cyberbullying bullying.*


Therefore, the hypothesis model of this study is shown in Figure 1.

## 2. Method

### 2.1. Research Design and Methods

#### 2.1.1. Interview Research Stage

The interview research was approved by the Academic Ethics Committee of Jing Hengyi School of Education of Hangzhou Normal University (Approval No: 2021002). During the interview study phase, an informed consent form for accepting the interview would be sent to the parents of the students one day in advance by the class teacher. After the parents confirmed their consent, the students were asked to be interviewed about non-physical bullying in school.

Twelve students (6 boys and 6 girls) from the fifth and sixth grades of elementary school were selected as participants. They included 4 participants randomly selected from the students with excellent comprehensive quality evaluation, 4 participants from the students with average comprehensive quality evaluation, and 4 participants from the students with comprehensive quality evaluation lower than the average. The interview questions included: Do any students in the class give nicknames to others? Do the students who are given nicknames approve of this behavior? Do you approve of this behavior? If this happened to you, would you feel bullied? Is there a student in your class that is isolated by other students? Do you know the reason why they were isolated? Are there any students in your class who post pictures of others who were scandalized in the WeChat group? How would you feel if these situations happened to you? These questions were used to examine the basic information of non-physical bullying-related incidents and behaviors that occurred in the upper elementary school years, as well as to assess the extent to which the interview results fit with the content of the research questionnaire used in this study. The presentation of the questionnaire was fine-tuned based on the interview results to suit the comprehension level of upper elementary school students [40]. Through the interviews, we found that there were big differences in the reported degrees of bullying and awareness of non-physical bullying events [50].

#### 2.1.2. Questionnaire Research Stage

In the questionnaire research phase, the survey was administered in class units. The informed consent form for filling out the questionnaire would be sent to the parents of the students one day in advance by the class teacher. After the parents confirmed their consent, the students were asked to anonymously fill in the questionnaires, ensuring that they understood the requirements. The filling-in time was 45 min in total, and all questionnaires were collected on the spot after the respondents had completed them. Following collection, the invalid questionnaires were eliminated and valid questionnaires were collated. This study was approved by the Academic Ethics Committee of Jing Hengyi Institute of Education of Hangzhou Normal University (Approval No: 2021002).

### 2.2. Sampling and Data Collection

Research data were collected from 8 primary schools in Zhejiang Province in 2021. The aims of the study were introduced to all respondents at the start guidelines of the questionnaire. The respondents had been told that their provided information would not be revealed to anyone and would solely be used for research purposes by the ethical rules of research. A randomized sampling survey method was used to control possible potential confounding factors. Anticipating a relatively lower response rate, a total of 560 questionnaires were distributed, and 492 questionnaires were received. Further analysis considered a total of 492 respondents, with an overall response rate of 87.86%. Details of the demographic data are shown in Table 1.

### 2.3. Measures

All instruments were psychometrically sound, as evidenced by the sufficient reliabilities of the scales in the current study.

#### 2.3.1. Negative Parenting Styles Scale

The Egna Minnen Beträffende Uppfostran Questionnaire (EMBU) was used to measure negative parenting styles. This questionnaire was originally developed by Perris et al. as one of the commonly used tools to measure parenting styles [51]. The standard version of the EMBU has a total of 81 questions, asking the children to recall the way their father and mother treated themselves when they were growing up and evaluate them from four levels: “never,” “occasionally,” “often,” and “always.” The questionnaire includes four dimensions: Rejection, Emotional Warmth, Overprotection, and Favoring Child. After that, Castro et al. revised the Egna Minnen Beträffende Uppfostran for Children (EMBU-C) for primary school children aged 7–12 [52]. The questionnaire has a total of 41 questions, including four dimensions “rejection,” “emotional warmth,” “overprotection,” and “preferring children.” Due to the good reliability and validity of the EMBU-C and its suitability for younger primary school children, the questionnaire has been widely used and revised worldwide [33,53]. Muris revised EMBU-C, deleted the dimension of “preferring children,” and added the dimension of “anxious rearing." The revised 40-item EMBU-C includes four dimensions, “Rejection,” “Emotional Warmth,” “Anxious Rearing,” and “Overprotection,” with good reliability and validity, widely used worldwide [34].

Considering the characteristics of the social situation in which students live and vertical development differences. The Egna Minnen Beträffende Uppfostran Questionnaire revised by Wang et al. in 2018 was used in this study [51,54]. The scale asks children to recall the way their fathers and mothers treated them when they were growing up, as well as to rate them on four levels: “never,” “occasionally,” “often,” and “always.” The questionnaire included three dimensions: rejection parenting style, overprotective parenting style, and anxious parenting style. Scale items include “Dad/Mum often tells me that he doesn’t like my behavior at home,” “Mom/Daddy treats me unfairly,” “Anything I do wrong will be punished by my dad/mum,” “I’ll be disappointed in dad/mum for not giving me what I want,” and “I feel like my dad/mom is mean and stingy towards me”. The questionnaire had good reliability, with alpha coefficients ranging from 0.51 to 0.86 for the three dimensions and split-half reliabilities ranging from 0.57 to 0.89.

#### 2.3.2. Negative Affect Experience Scale

The Negative Affect Experience Scale, prepared by Watson and revised by Zheng and Wang [35,55], contains a total of nine items. Items are rated with 5-point optional responses ranging from none or very slight (1) to very strong (5). Participants were asked to rate the negative emotional experiences they had felt when they were nicknamed, isolated, or posted embarrassing photos on WeChat by others since the start of school. Scale items include “scared,” “ashamed,” “angry.” The scale has sufficient reliability of 0.84.

#### 2.3.3. Negative Coping Style Scale

The Negative Coping Style Scale of the Xie Brief Coping Style Scale, which consists of 8 items, was used [56]. It has four options on a rating scale ranging from “don’t use” (0) to “often use” (3). Scale items include “trying to forget the whole thing," “Fantasy that some kind of miracle might happen to change the status quo,” and “Attempt to rest or take physical leave to temporarily put the problem (annoyance) aside.” The internal consistency of the scale was satisfactory (α = 0.78).

#### 2.3.4. Delaware Bullying Victimization Scale–Student

A portion of the Delaware Bullying Victimization Scale–Student (DBVS-S) was selected as the instrument [57], with 12 items divided into three dimensions: perceived verbal bullying (4 items), perceived relational bullying (4 items), and perceived cyberbullying (4 items). It had 6 options on the rating scale ranging from never (0) to every day (5). Scale items include “Others call me bad nicknames, or make fun of and ridicule me” “Some classmates posted my ugly photos online,” “Classmates told or urged others not to be friends with me.” The internal consistency of the scale was satisfactory (α = 0.839).

#### 2.3.5. Demographic Information

Based on the researcher’s long-term longitudinal observation of primary school students and the previous empirical research results on bullying, this study identified demographic variables such as gender [58,59,60,61,62,63], grade [64,65], whether or not to be an only child [66,67,68], and integrated and utilized them as confounding variables in the statistical control of this study. In previous studies, the age of the respondents was a characteristic that was easily confounded in the field of bullying research. In this study, considering that the current primary school children in China are generally enrolled at the right age, they all graduated as scheduled after six years of primary school, regardless of their grades and performance, and that there is no repetition of grades in any case, so students in grade five or six are selected as the research objects, and the age and age characteristics of the respondents are reflected by grades. In the process of this questionnaire survey, the information of this part of the research object is collected by setting the corresponding demographic options in the questionnaire, allowing students to choose the corresponding answers according to their actual situation.

### 2.4. Statistical Analysis

All questionnaire data were processed and analyzed with SPSS26.0 and Amos 20.0. The results of the questionnaire were analyzed using statistical methods, such as Pearson correlation analysis, the independent sample *t*-test, multiple hierarchical regression, and structural equation modeling.

## 3. Results

The data from the filled survey questionnaires were entered into the SPSS file. The data with outliers were deleted in the whole line, and the missing values were processed by the method of mean substitution.

### 3.1. Difference Analysis

Table 2 presents the means and standard deviations of all scales used in the present study. The results indicated that the perceived verbal bullying significantly differed according to gender, grade, and only/non-only children (*p* < 0.05). Male, fifth grade, and non-only child students presented higher scores of the perceived verbal bullying than female, sixth grade, and only child students. The perceived relationship with bullying significantly differed between grades (*p* < 0.05). The gender difference in the perceived cyberbullying also reached a significant level (*p* < 0.05), where boys’ perceived cyberbullying scores were higher than those of girls.

As seen in Table 2, there were significant differences in the negative affect experiences of upper elementary school grade students and only/non-only children (*p* < 0.05). The negative affect experience scores of the sixth-grade students were significantly lower than those of the fifth-grade students, and the negative affect experience scores of only children were significantly lower than those of non-only children. However, the differences in the negative coping styles of the upper primary school grade students in terms of gender, grade, and only/non-only children did not reach statistically significant levels (*p* > 0.05).

The results also showed that the differences in anxious parenting styles according to gender and grade did not reach statistically significant levels, while the difference in only/non-only children was statistically significant (*p* < 0.05). Furthermore, the anxious parenting style scores of students with siblings were significantly higher than those of students with only children. The differences in the rejection parenting style according to grades and only/non-only children reached a statistically significant level (*p* < 0.05), and the gender difference regarding the overprotective parenting style reached a significant level (*p* < 0.05).

### 3.2. Descriptive Statistics and Correlation Analysis

Descriptive analyses were conducted and correlations between the variables were examined (Table 3). As seen in Table 3, the perceived cyberbullying and anxious parenting styles were not significantly correlated, while the other variables were all significantly positively correlated (*p* < 0.05).

### 3.3. Multiple Hierarchical Regression Analysis

Multiple hierarchical regression analysis was performed using SPSS.26.0 to examine the variables’ respective contributions in explaining the dependent variable. The purpose of the analysis was to look at the true contribution of the independent variable to the dependent variable after excluding potential confounding factors. Table 4 shows the results of the multiple hierarchical regression analysis. First, hierarchical regression analysis was performed with verbal bullying as the dependent variable. In the first step, the control variable is introduced into the regression equation to construct model 1; in the second step, the negative parenting styles are also introduced into the regression equation to construct model 2; in the third step, negative affect experience and negative coping style are introduced into the equation, and model 3 is constructed. Model 1 indicated that 3% of the variance of the perceived verbal bullying could be attributed to gender and only child (F = 5.086, *p* < 0.01). Among them, gender has a significant negative predictive effect on verbal bullying (β = −0.757, *p* < 0.05), and only child has a significant positive predictive effect on verbal bullying (β = 1.085, *p* < 0.01). Model 2 indicated that 9.2% of the variance of the perceived verbal bullying could be attributed to the rejection parenting style (F = 11.177, *p* < 0.001). And rejection parenting style has a significant positive predictive effect on verbal bullying (β = 0.143, *p* < 0.001). Model 3 indicated that 13.3% of the variance of the perceived verbal bullying could be attributed to negative affect experience and negative coping style (F = 20.637, *p* < 0.001). Negative affect experience can significantly and positively predict verbal bullying (β = 0.227, *p* < 0.001). Negative coping style can significantly and positively predict verbal bullying (β = 0.097, *p* < 0.05).

Second, multiple hierarchical regression analysis was performed with cyberbullying as the dependent variable. Table 4 shows the results. In the first step, the control variable is introduced into the regression equation to construct model 1; in the second step, the negative parenting style is also introduced into the regression equation to construct model 2; in the third step, negative affect experience and negative coping style are introduced into the equation, and model 3 is constructed. Model 1 indicated that 1.6% of the variance of the perceived cyberbullying could be attributed to gender (F = 2.646, *p* < 0.05). Gender has a significant negative predictive effect on cyberbullying (β = −0.570, *p* < 0.05). Model 2 indicated that 4.1% of the variance of the perceived cyberbullying could be attributed to the rejection parenting style of the negative parenting style (F = 4.874, *p* < 0.001). Rejection parenting style has a significant positive predictive effect on cyberbullying (β = 0.068, *p* < 0.001). Model 3 indicated that 6.1% of the variance of the perceived cyberbullying could be attributed to negative affect experience (F = 8.082, *p* < 0.001). Negative affect experience can positively predict cyberbullying (β = 0.123, *p* < 0.001) significantly.

Third, multiple hierarchical regression analysis was performed with relationship bullying as the dependent variable. Table 4 shows the results. In the first step, the control variable is introduced into the regression equation to construct model 1; in the second step, the negative parenting style is also introduced into the regression equation to construct model 2; in the third step, negative affect experience and negative coping style are introduced into the equation, and model 3 is constructed. Model 1 indicated that 1.2% of the variance of the perceived relationship bullying could be attributed to gender, grade, and only child (F = 1.950, *p* > 0.05), but the regression coefficients of all predicted variables are not significant. The regression equation is not valid. Model 2 indicated that 10.5% of the variance of the perceived relationship bullying could be attributed to the rejection parenting style of the negative parenting style (F = 10.642, *p* < 0.001). Rejection parenting style has a significant positive predictive effect on cyberbullying (β = 0.123, *p* < 0.001). Model 3 indicated that 8.7% of the variance of the perceived relationship bullying could be attributed to negative affect experience (F = 15.432, *p* < 0.001). Negative affect experience can significantly and positively predict cyberbullying (β = 0.167, *p* < 0.001).

Finally, multiple hierarchical regression analysis was performed with non-physical bullying total score as the dependent variable. In the first step, the control variable is introduced into the regression equation to construct model 1; in the second step, the negative parenting style is also introduced into the regression equation to construct model 2; in the third step, negative affect experience and negative coping style are introduced into the equation, and model 3 is constructed. Model 1 indicated that 2.1% of the variance of the perceived non-physical bullying could be attributed to gender and only child (F = 3.448, *p* < 0.05). Among them, gender has a significant negative predictive effect on non-physical bullying total score (β = −0.146, *p* < 0.05), and only child has a significant positive predictive effect on non-physical bullying total score (β = 0.148, *p* < 0.05). Model 2 indicated that 9.6% of the variance of the perceived non-physical bullying could be attributed to the rejection parenting style of the negative parenting style (F = 10.719, *p* < 0.001). Rejection parenting style has a significant positive predictive effect on non-physical bullying total score (β = 0.016, *p* < 0.001). Model 3 indicated that 11.3% of the variance of the perceived non-physical bullying could be attributed to negative affect experience (F = 20.643, *p* < 0.001). Negative affect experience can significantly and positively predict non-physical bullying total score (β = 0.043, *p* < 0.001).

### 3.4. Mediation Analysis

Based on the literature research and the results of multiple hierarchical regression analysis in Table 4, a structural equation model of the relationship between rejection parenting style, negative affective experience, negative coping style, and perceived verbal bullying was constructed. Figure 2 presents the test results of the influence on the perceived verbal bullying in the upper-grade primary school students, with the rejection parenting style as an independent variable and negative affect experience and negative coping styles as mediating variables. Table 5 shows the model goodness-of-fit indices. Among them, CMIN/DF = 2.052, RMSEA = 0.046, CFI = 0.903, and GFI = 0.922 were all within their acceptable ranges, which indicated that the model fit well and that negative affect experience and negative coping styles presented a chain-like mediating effect between the rejection parenting style and the perceived verbal bullying in the upper-grade primary school students.

Based on the literature research and the results of multiple hierarchical regression analysis in Table 4, a structural equation model of the relationship between rejection parenting style, negative affective experience, and perceived cyberbullying was constructed. A mediation effect test model was shown (in Figure 3), with the perceived cyberbullying as the dependent variable, the rejection parenting style as an independent variable, and the negative affect experience as a mediating variable. Table 6 shows the model goodness-of-fit indices. Among them, CMIN/DF = 2.515, RMSEA = 0.056, CFI = 0.934, and GFI = 0.940, which were all within their acceptable ranges; this indicated that the model fit well. It was shown that negative affect experience had a partial mediating effect between the rejection parenting style and the perceived cyberbullying in the upper-grade elementary school students.

Based on the literature research and the results of multiple hierarchical regression analysis in Table 4, a structural equation model of the relationship between rejection parenting style, negative affective experience, and perceived relationship bullying was constructed. A mediation effect test model was shown (in Figure 4), with the perceived relationship bullying as the dependent variable, the rejection parenting style as an independent variable, and negative affect experience as a mediating variable; Table 7 shows the model goodness-of-fit indices. Among them, CMIN/DF = 2.523, RMSEA = 0.056, CFI = 0.936, and GFI = 0.940, which were all within their acceptable ranges; this indicated that the model fit well. It was shown that negative affect experience had a partial mediating effect between the rejection parenting style and the perceived relationship bullying in the upper-grade elementary school students.

Based on the literature research and the results of multiple hierarchical regression analysis in Table 4, a structural equation model of the relationship between rejection parenting style, negative affective experience, and perceived non-physical bullying total score was constructed. A mediation effect test model was shown (in Figure 5), with the perceived non-physical bullying total score as the dependent variable, the rejection parenting style as the independent variable, and the negative affect experience as the mediating variable. Table 8 shows the model goodness-of-fit indices. Among them, CMIN/DF = 2.132, RMSEA = 0.048, CFI = 0.916, and GFI = 0.922, which were all within the acceptable range; this indicated that the model fit well. It was shown that negative affect experience had a partial mediating effect between the rejection parenting style and the perceived school non-physical bullying in the upper-grade elementary school students.

## 4. Discussion

This study aimed to explore the impact of parenting styles on the students’ perceived school non-physical bullying and the effects of negative affect experiences and negative coping styles between them. Correlation analysis shows that negative affect experiences, negative coping styles, negative parenting styles, and the perceived school non-physical bullying are all positively related to each other. These findings are consistent with those of recent studies [50,69,70]. Wójcik & Rzeńca (2021) found a significant link between negative coping and being bullied [50]. Other research reported that bullying victim recognition was significantly correlated with the degree of the parent–child relationship (Zhao et al., 2021) [69]. Rauschenberg et al. (2020) found that bullied people reported extremely strong negative emotional experiences [70]. These findings confirmed hypotheses 1, 2, and 4.

The results of multiple hierarchical regression analysis showed that, after controlling the confounding variables such as gender, grade, and only child or not, rejection parenting style, negative emotional experience, and negative coping style still had a significant influence on students’ perceived school non-physical bullying.

The results of mediating effect analysis by structural equation modeling showed that negative affect experiences and negative coping styles had a chain-like mediating effect between the rejection parenting style and students’ perceived school verbal bullying. The findings confirmed hypothesis 6. Moreover, negative affect experiences had a partial mediating effect between the rejection parenting style and students’ perceived school non-physical bullying total scores, relationship bullying, and cyberbullying [69,71]. These findings partially confirmed hypotheses 3, 7, and 8. In this study, only hypothesis 5 has not been confirmed.

### 4.1. Correlation and Regression Analysis of Parenting Styles and the Perceived School Non-Physical Bullying

The results of the present study demonstrated significant correlations between negative parenting styles, negative affect experiences, negative coping styles, and perceived school non-physical bullying in senior primary school students. We found significant positive correlations between the anxiety, rejection, overprotective parenting styles; negative emotional experiences; and negative coping styles. This was consistent with the research finding of Wu & Liu (2009), showing that fathers’ rejection and overprotective parenting styles may cause students to adopt negative coping styles to deal with problems [72]. Negative affect experiences and negative coping styles were shown to be significantly positively correlated with students’ perceived verbal bullying, relationship bullying, and cyberbullying. A similar conclusion was reached by researcher Song (2018), who reported that psychological bullying (including verbal bullying, relationship bullying, and cyberbullying) perceived by students is positively correlated with negative coping styles [73]. Among the three forms of perceived school non-physical bullying, only the perceived cyberbullying showed an insignificant correlation with anxious parenting styles. Perceived verbal bullying and relationship bullying were shown to be significantly positively correlated with the overprotective, rejection, and anxious parenting styles. In related studies, parents’ overprotective and mothers’ rejection parenting styles were found to have significantly affected junior high school students’ perceived relationship and verbal bullying in the context of school non-physical bullying [74]. Other studies pointed out that parents’ overprotective and rejection parenting styles are significantly positively correlated with junior high school students’ perceived cyberbullying [75]. These research conclusions were consistent with the results of the present study, indicating that the psychological development characteristics of senior primary school students in this regard are at the same level as those of junior middle school students.

According to the results of multiple hierarchical regression analysis, rejection parenting, negative affect experiences, and negative coping styles jointly explained 22.5% of the total variation of perceived verbal bullying (F = 20.637, *p* < 0.01); the rejection parenting style and negative affect experiences jointly explained 19.2% of the total variation of perceived social relationship bullying (F = 15.432, *p* < 0.01); rejection parenting style and negative affect experiences jointly explained 10.2% of the total variation of perceived cyberbullying (F = 8.082, *p* < 0.01); and negative affect experience and the rejection parenting style jointly explained 21.1% of the total variation of the total score of perceived non-physical bullying (F = 18. 162, *p* < 0.01). These results demonstrate that the rejection parenting style was an important factor that affected the students’ perceived school non-physical bullying. Parents’ blind criticism or rejection, such as “unreasonable reprimand and punishment” and “unfair and petty treatment,” make children more likely to perceive non-physical bullying at school. One possible explanation is that parents’ blind rejection and denial in the family environment comprise a situation that is similar to the non-physical verbal, relationship, and cyberbullying experienced by students at school. The mentioned behaviors of parents harm children’s self-esteem and self-confidence, thus leading to self-recognition difficulties and the acceptance of non-physical bullying by others [76]. Negative emotions, such as derogation and unfair treatment previously experienced by students in the family environment, can transfer to experiences similar to school non-physical bullying, and the negative coping styles adopted under the rejection parenting style can transfer to similar coping styles of school non-physical bullying. As a result, students who grow up under the rejection parenting style are more likely to perceive school non-physical bullying.

### 4.2. The Mediating Effect of Negative Affect Experience and Negative Coping Styles between the Rejection Parenting Style and the Perceived School Non-Physical Bullying

The results of the mediating effect test model showed that negative emotional experiences and negative coping styles had a significant chain mediating effect between the rejection parenting style and the students’ perceptions of school verbal bullying, and negative emotional experiences significantly positively predicted the perceived school verbal bullying by influencing the negative coping styles. These results showed that the rejection parenting style not only directly affected the students’ perceived school verbal bullying but also indirectly affected the students’ perceived school verbal bullying by influencing negative emotional experiences and negative coping styles.

During the process of a student’s growth, parents frequently reject, deny, and adopt negative attitudes toward them in all aspects of study and life, e.g., “disappointment,” “unfairness,” “pettiness,” and “punished for no reason.” As such, students are placed in a negative family atmosphere and growth environment for a long time, which makes them cold and indifferent, thus generating the perception that they are not cared for by their families. Over time, students grow to feel insecure, thus intensifying their negative affect experiences [77]. When facing problems, they are prone to having negative affect experiences, such as “tension” and “irritability,” which may lead them to adopt simple and negative coping styles when solving problems [78,79]. This kind of rejection parenting style had negative impacts on students’ affect experiences and coping styles. Our analysis showed that the degrees of negative affect experiences and negative coping styles were correlated with the students’ perceptions of school non-physical bullying, and the rejection parenting style was observed to affect students’ perceived school non-physical bullying by influencing students’ negative affect experiences and negative coping styles.

The partial mediating effect of negative affect experiences between the rejection parenting style and the total score of students’ perceptions of school non-physical bullying, relationship bullying, and cyberbullying suggested that the rejection parenting style not only directly affected students’ perceived school relationship bullying and cyberbullying but also indirectly impacted students’ perceived school relationship bullying and cyberbullying by influencing their negative affect experiences.

The results of the above-mentioned mediating effect analyses indicate that there were differences in the psychological mechanisms of action of the rejection parenting style on students’ perceived school verbal bullying, relationship bullying, and cyberbullying.

### 4.3. Theoretical Implications

In previous bullying studies, the association between the rejection parenting style and perceived school non-physical bullying as well as the mediation effects of negative affect experiences and cope in this association is limited. This research highlights the significant influence of rejection parenting style on students’ perceived school non-physical bullying behavior. The nature of the emotional-behavioral bond between parents and children has an important impact on children’s interpretation of the behavior of others. This suggests that children in the context of the rejection parenting style are more likely to establish a link between others’ non-physical bullying behaviors and their self-perceived non-physical bullying.

In addition, this study reveals the mediating role of negative affective experiences and coping in students’ perceived school non-physical bullying in the context of family rejection parenting. The mediating effect of negative affective experiences and coping shows that the rejection parenting styles affect negative affect experiences, and negative affect experiences also easily lead to negative coping styles. Besides, the effect of rejection parenting style on perceived school non-physical bullying may depend on the level of negative affect experiences and coping, especially in students’ perceived school verbal bullying.

This study enriches the related research results of non-physical bullying in the past from the perspective of refusing parenting style, negative affect experiences and coping. It also provides a theoretical basis and empirical data support for the subsequent related research on non-physical bullying.

## 5. Practical Implications

### 5.1. Schools

The outcomes of this research suggest some practical implications for schools. Non-physical bullying mainly occurs at schools. Therefore, corresponding measures taken by the schools play important roles in students’ perceived school non-physical bullying.

First, schools should use parents’ meetings to guide students’ parents to realize the impact of family education on students’ perceived school non-physical bullying, and they should advise parents to rarely, if ever, use the rejection parenting style at home. Second, after students perceive a school non-physical bullying incident, the school should take corresponding measures to prevent the bullying from continuing and carry out effective interventions to minimize bullied students’ negative emotional experiences, such as paying full attention to the students who are aware of school non-physical bullying, giving the bullied timely and enough guidance and help, and providing timely and effective psychological counseling, support, and comfort.

### 5.2. Parents

In this study, it was found that the rejection parenting style is an important factor that affected students’ perceptions of school non-physical bullying because students’ negative affect experiences under the rejection parenting style were easily transferred to experiences that were similar to school non-physical bullying. The negative coping styles adopted by students under the rejection parenting style also easily migrated to coping styles similar to school verbal bullying. Therefore, students growing up under the rejection parenting style were more likely to perceive school non-physical bullying. Accordingly, parents should try their best to be cautious with or avoid using the rejection parenting style to educate their children. In addition, to help children reduce the adverse effects of non-physical bullying, parents should also do the following:

Firstly, parents should help their children cope with negative emotional experiences. In daily family education, parents should pay attention to their children’s emotional expressions, identify their abnormal emotions in time, and take appropriate and effective measures to provide their children with sufficient companionship, security, love, and care to prevent them from having overly negative school and peer emotional experiences. It is also important for parents to guide their children to release negative emotions and restore calm and positive emotional experiences. If the child’s emotional problems are serious enough, parents should take their children to seek the help of psychological professionals, schools, and teachers to help children vent extreme negative emotions in time and avoid extreme behavior.

Secondly, parents should teach their children positive coping styles. For example, they should help children calmly face non-physical bullying, have full confidence, and build sufficient and effective coping strategies to resolve difficulties. When students observe the school’s non-physical bullying of another student, they should be able to provide appropriate and timely help, e.g., stop it or notify teachers, so that relevant non-physical bullying events at school can be actively and properly handled.

## 6. Limitations and Future Research

Based on the theoretical basis of previous research, this paper studies and analyzes the relationship between the parenting style of fifth and sixth-grade students and the perceived school non-physical bullying using interviews and questionnaires. Moreover, it discusses the role of the negative affect experiences and negative coping styles in the above relationship. The topic selection has certain research innovations, but due to the limitations of practice and theory, there are still some deficiencies that need to be further improved by follow-up research.

Firstly, this research mainly adopts the questionnaire survey method, which may be interfered with some unclear factors, so the results of this research may be affected to a certain extent. Subsequent research can use more abundant and rigorous research methods to explore, for example, pairing the bully and the bullied to explore the reasons for the differences in the behaviors emitted and felt by two different children in the same behavior, thereby improving the accuracy of research results.

Secondly, from the perspective of the research object, the object of this study is the fifth and sixth-grade students of 8 primary schools in Zhejiang Province. Although the sample is representative to a certain extent, the generality of the research results needs to be further tested in the follow-up research. To improve the representativeness of research sampling and further verify the results of this study, future research can expand the scope of sample selection and select sample groups from different regions and cities.

Thirdly, the rejection parenting style might also impact children’s self-concept and make them act in ways in which others perceive them differently and make them more likely to be a target of bullying. This is also the direction that future research needs to verify further.

## Figures and Tables

**Figure 1 ijerph-19-06206-f001:**
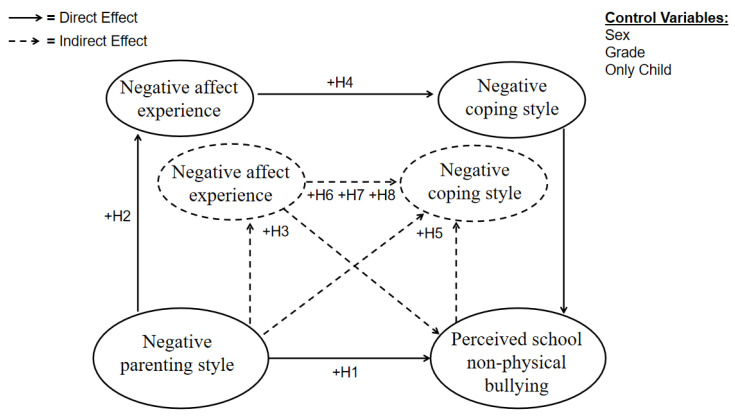
Hypothesized model.

**Figure 2 ijerph-19-06206-f002:**
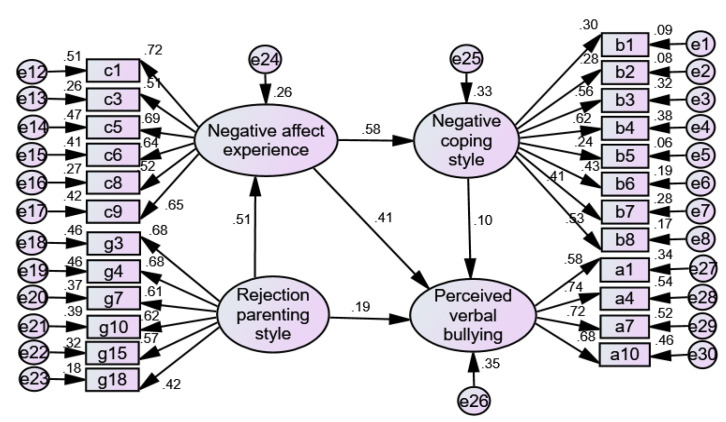
The chain-like mediating effect of negative affect experience and negative coping styles between rejection parenting style and the perceived verbal bullying.

**Figure 3 ijerph-19-06206-f003:**
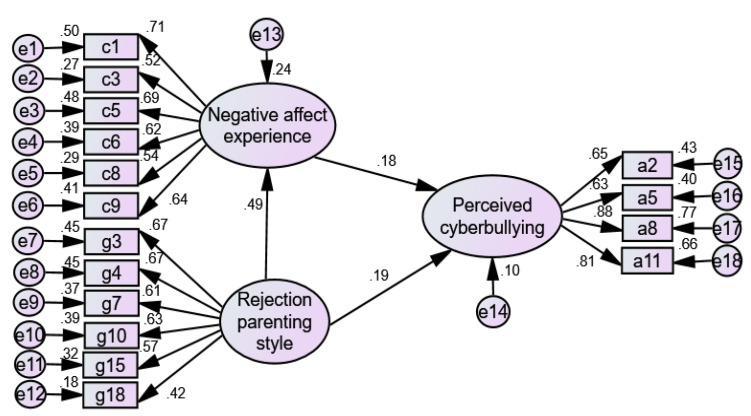
Partial mediating effect of negative affect experience between the rejection parenting style and the perceived cyberbullying.

**Figure 4 ijerph-19-06206-f004:**
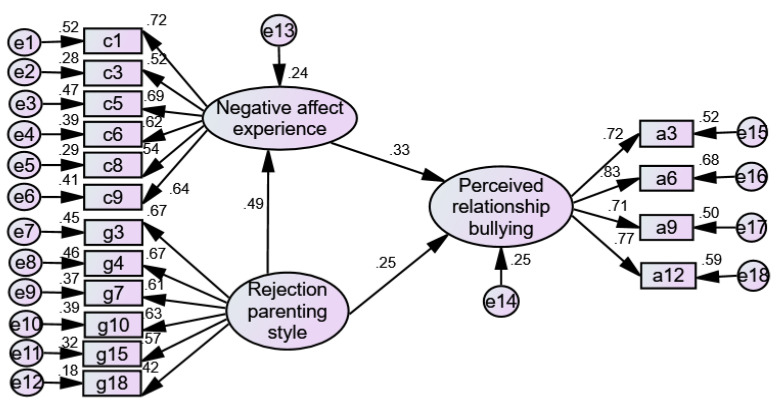
Partial mediating effect of negative affect experience between the rejection parenting style and the perceived relationship bullying.

**Figure 5 ijerph-19-06206-f005:**
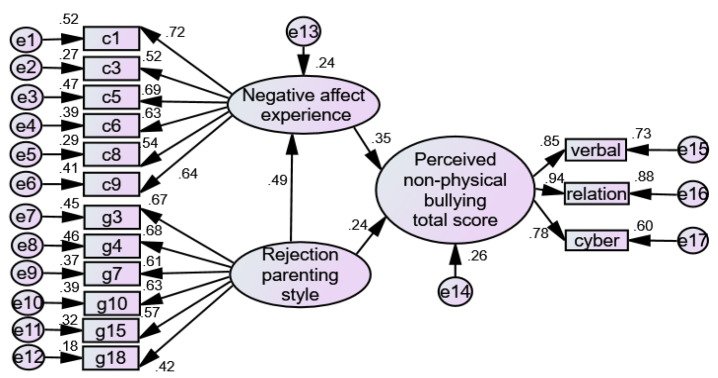
Partial mediating effect of negative affect experience between the rejection parenting style and the perceived non-physical bullying on school.

**Table 1 ijerph-19-06206-t001:** Sample characteristics.

Measure	Items	Frequency (n)	Percentage (%)
Gender	Male	247	50.20
	Female	245	49.80
Grade	Fifth	222	45.12
	Sixth	270	54.88
Only/Non-Only Children	Yes	174	35.37
	No	318	64.63

**Table 2 ijerph-19-06206-t002:** Difference analysis of the scale scores of the present study (N = 492).

Group		Statistics	Negative Affect Experience	Parenting Style			Non-Physical Bullying in School		
				Anxious	Rejection	Overprotective	Verbal	Relationship	Cyber
Gender	Boy		0.893 ± 0.3796	3.532 ± 1.139	2.562 ± 0.895	3.286 ± 1.097	0.763 ± 1.049	0.417 ± 0.901	0.270 ± 0.789
	Girl		0.897 ± 0.363	3.389 ± 1.061	2.451 ± 0.772	3.057 ± 0.961	0.586 ± 0.810	0.309 ± 0.624	0.138 ± 0.418
		*t*	−0.126	1.443	1.473	2.462 *	2.101 *	1.544	2.330
		*p*	0.900	0.150	0.141	0.014	0.036	0.123	0.020 *
Grade	Fifth		0.946 ± 0.369	3.515 ± 1.124	2.603 ± 0.861	3.234 ± 1.059	0.793 ± 0.944	0.451 ± 0.789	0.205 ± 0.574
	Sixth		0.840 ± 0.362	3.393 ± 1.068	2.400 ± 0.770	3.086 ± 0.952	0.525 ± 0.834	0.267 ± 0.689	0.186 ± 0.625
		*t*	3.157 **	1.203	2.687 **	1.603	3.261 **	2.685 **	0.347
		*p*	0.002	0.230	0.007	0.110	0.001	0.008	0.729
Only child	Yes		0.783 ± 0.310	3.297 ± 1.014	2.300 ± 0.638	3.157 ± 0.995	0.504 ± 0.748	0.300 ± 0.731	0.160 ± 0.617
	No		0.957 ± 0.388	3.556 ± 1.137	2.622 ± 0.910	3.182 ± 1.061	0.767 ± 1.022	0.398 ± 0.801	0.230 ± 0.645
		*t*	−5.423 **	−2.504 *	−4.566 **	−0.259	−3.251 **	−1.337	−1.168
		*p*	0.000	0.013	0.000	0.796	0.001	0.182	0.243

* *p* < 0.05, ** *p* < 0.01.

**Table 3 ijerph-19-06206-t003:** Descriptive statistics and correlation matrix for all variables (N = 492).

Variables	M	SD	1	2	3	4	5	6	7
1. Anxious parenting style	20.770	6.612							
2. Rejection parenting style	22.560	7.520	0.362 **						
3. Overprotective parenting style	19.030	6.221	0.642 **	0.543 **					
4. Negative affect experience	14.330	5.939	0.210 **	0.400 **	0.275 **				
5. Negative coping style	6.200	3.865	0.160 **	0.278 **	0.216 **	0.409 **			
6. Verbal bullying	2.700	3.764	0.095 *	0.324 **	0.193 **	0.461 **	0.280 **		
7. Relationship bullying	1.450	3.106	0.090 *	0.328 **	0.210 **	0.399 **	0.201 **	0.795 *	
8. Cyberbullying	0.820	2.539	0.050	0.210 *	0.119 **	0.302 **	0.106 *	0.655 **	0.736 **

* *p* < 0.05, ** *p* < 0.01.

**Table 4 ijerph-19-06206-t004:** Multiple hierarchical regression analysis of the perceived non-physical bullying on negative parenting style, negative affect experience, and negative coping styles (N = 492).

Dependent Variable	Independent Variable	β	R^2^	Adjust R^2^	F
Verbal bullying	Model 1		0.030	0.024	5.086 **
	Gender	−0.757 *			
	Grade	−0.207			
	Only child	1.085 **			
	Model 2		0.122	0.111	11.177 ***
	Gender	−0.577			
	Grade	−0.121			
	Only child	0.735 *			
	Anxious parenting style	−0.039			
	Rejection parenting style	0.143 ***			
	Over-protection	0.045			
	Model 3		0.255	0.243	20.637 ***
	Gender	−0.703 *			
	Grade	−0.090			
	Only child	0.270			
	Anxious parenting style	−0.046			
	Rejection parenting style	0.076 **			
	Over-protection	0.019			
	Negative affect experience	0.227 ***			
	Negative coping style	0.097 *			
Cyberbullying	Model 1		0.016	0.010	2.646 *
	Gender	−0.570 *			
	Grade	0.192			
	Only child	0.324			
	Model 2		0.057	0.045	4.874 ***
	Gender	−0.496 *			
	Grade	0.232			
	Only child	0.161			
	Anxious parenting style	−0.020			
	Rejection parenting style	0.068 ***			
	Over-protection	0.014			
	Model 3		0.118	0.104	8.082 ***
	Gender	−0.531 *			
	Grade	0.305			
	Only child	−0.069			
	Anxious parenting style	−0.023			
	Rejection parenting style	0.041 *			
	Over-protection	0.004			
	Negative affect experience	0.123 ***			
	Negative coping style	−0.026			
Relationship bullying	Model 1		0.012	0.006	1.950
	Gender	−0.427			
	Grade	−0.299			
	Only child	0.371			
	Model 2		0.117	0.106	10.642 ***
	Gender	−0.264			
	Grade	−0.225			
	Only child	0.077			
	Anxious parenting style	−0.039			
	Rejection parenting style	0.123 ***			
	Over-protection	0.049			
	Model 3		0.204	0.191	15.432 ***
	Gender	−0.335			
	Grade	−0.165			
	Only child	−0.250			
	Anxious parenting style	−0.043			
	Rejection parenting style	0.079 ***			
	Over-protection	0.032			
	Negative affect experience	0.167 ***			
	Negative coping style	0.020			
Non-physical bullying total score	Model 1		0.021	0.015	3.448 *
	Gender	−0.146 *			
	Grade	−0.026			
	Only child	0.148 *			
	Model 2		0.117	0.106	10.719 ***
	Gender	−0.111			
	Grade	−0.010			
	Only child	0.081			
	Anxious parenting style	−0.008			
	Rejection parenting style	0.028 ***			
	Over-protection	0.009			
	Model 3		0.232	0.219	18.162 ***
	Gender	−0.131 *			
	Grade	0.004			
	Only child	−0.004			
	Anxious parenting style	−0.009			
	Rejection parenting style	0.016 ***			
	Over-protection	0.005			
	Negative affect experience	0.043 ***			
	Negative coping style	0.008			

* *p* < 0.05, ** *p* < 0.01, *** *p* < 0.001.

**Table 5 ijerph-19-06206-t005:** The model fitting goodness index table with the perceived verbal bullying as the dependent variable.

Goodness-of-Fit Indices	CMIN/DF	CFI	IFI	GFI	AGFI	RMSEA
Result	2.052	0.903	0.904	0.922	0.905	0.046

**Table 6 ijerph-19-06206-t006:** The model fitting goodness index table with the perceived cyberbullying as the dependent variable.

Goodness-of-Fit Indices	CMIN/DF	CFI	IFI	GFI	AGFI	RMSEA
Result	2.515	0.934	0.935	0.919	0.940	0.056

**Table 7 ijerph-19-06206-t007:** The model fitting goodness index table with the perceived relationship bullying as the dependent variable.

Goodness-of-Fit Indices	CMIN/DF	CFI	IFI	GFI	AGFI	RMSEA
Result	2.523	0.936	0.936	0.919	0.940	0.056

**Table 8 ijerph-19-06206-t008:** The model fitting goodness index table with the perceived non-physical bullying total score as the dependent variable.

Goodness-of-Fit indices	CMIN/DF	CFI	IFI	GFI	AGFI	RMSEA
Result	2.132	0.916	0.917	0.904	0.922	0.048

## Data Availability

According to the data access policies, the data used to support the findings of this study are available from Hangzhou Normal University upon a reasonable request made by email: zhouhouyu1@hznu.edu.cn.

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
