# Peer review of "Negative Parenting Style and Perceived Non-Physical Bullying at School: The Mediating Role of Negative Affect Experiences and Coping Styles"

_ijerph, 2022, doi:10.3390/ijerph19106206_

Round 1
Reviewer 1 Report
Major concern: The revised title doesn't make sense and is not written in proper English grammar. Please revise.
Generally, please have a native English speaker review the entire manuscript. There are many areas of awkward wording, incorrect grammar (e.g., missing articles), and unclear sentences, e.g., "At present, school bullying incidents occur frequently, which attracts increasing attention of researchers" in the Abstract should be something like, "At present, school bullying incidents occur frequently, attracting increased attention from researchers."
Abstract:
Point 1 - Delete "by using the questionnaires" after "were surveyed," as this is redundant (line 16). Add a comma and an article in the following: "For controlling any potential confounding factors, a randomized sampling survey method was used to distribute questionnaires." On line 24, "it was observed to be directly associated with students’ perceived school non-physical bullying" (add the word "be"). ETC. -
Introduction -
The Introduction is too long, going into excessive detail for every construct. It should be edited down and include only major references that are the most recent.
Point 1 - As noted in the first review, using the term "victim" is pejorative for targets of bullying. The authors would like to continue to use this term, and state: "The victims here include
target of bullying and the victim. For brevity, we will use victim instead of both in our article" on lines 41-42. This doesn't make sense and does not capture the point. Instead, if the authors and editors decide to continue to use the term "victim," they can state something like, ""Victim" will be used throughout this manuscript rather than "target of bullying" for brevity's sake."
Point 2 - This sentence does not make any sense. Please reword to make the meaning more clear: (lines 64-65) "Studies have shown the link between the Perceived verbal bullying and self-harm behavior: the perceived verbal bullying in non-physical bullying greatly increases the possibility of self-harm among young people."
Point 3 - Would recommend not using the phrase "emotionally disturbed" (not appropriate for a study of children) on lines 70-71 - use more precise language, such as experience higher levels of depression and symptoms of anxiety, or whatever was found in the cited study.
Point 4: This sentence doesn't make sense - perceived non-physical bullying on school? (page 3, lines 98-99) Please reword - "The family parenting style is an important factor that influences the occurrence of non-physical bullying at school."
Point 5 - "hyperactive-attention deficits" isn't an appropriate term to characterize ADHD - instead, could state "...correlated with hyperactivity and attention deficits" on page 3, lines 107-108)
Point 6 - Another example of awkward wording: "thus are more likely to perceived non-physical bullying on school" - perhaps the authors mean "thus are more likely to perceive that they are being the target of non-physical bullying at school." (page 3, lines 115-116).
Methods -
Point 1 - Do not start a sentence with a number (12 students).
Point 2 - What constituted an "invalid questionnaire"? How many (what percentage) were deemed invalid?
Point 3 - Recommend that the authors do not use the word "subjects" on page 6 line 283, but instead use the word "children." "Subjects" is more appropriate to animals in experimental research.
Point 4 - For the Negative Affect Experience Scale, what is the time period over which the participants are rating experiencing the negative emotions?
Discussion -
Point 1 - In my first review, I mentioned not using causative words like "impact," as this is a cross-sectional study. However, the word impact still appears throughout the manuscript, including in the first sentence of the Discussion. Please search for this word and add in something like "association" or "relationships" - i.e., "to explore the relationships of parenting styles on the students' perceived school non-physical bullying..." All of these constructs are being measured at the same time, so causation cannot be implied, correct? It's also not appropriate to say that one variable had a "significant influence" on another (see line 15, line 533). Instead, the variables were related, associated, or correlated with each other.
Point 2 - The Discussion is also extremely long and would benefit from a more concise presentation to read like a research article.
References - While the authors believe that including dissertations is justified because there are experts on the committees, there is a major problem with potential bias and conflict of interest. The members of dissertation committees are oftentimes at the same program and institution and some are even in the same lab or research group. The expert committee chair and members have a vested interest in having the person defending the dissertation pass and succeed. I will defer to the Editors, but I would recommend only citing research studies that have been published in scientific journals and have gone through a review of peers that is unbiased and free from conflicts of interest.
Author Response
Response to Reviewer 1 Comments
Comments and Suggestions for Authors
Point 1: Major concern: The revised title doesn't make sense and is not written in proper English grammar. Please revise.
Response 1:
Rejection Parenting Style and Perceived Non-physical Bullying at School: The Mediating Role of Negative Affect Experiences and Coping Styles
Point 2: Generally, please have a native English speaker review the entire manuscript. There are many areas of awkward wording, incorrect grammar (e.g., missing articles), and unclear sentences, e.g., "At present, school bullying incidents occur frequently, which attracts increasing attention of researchers" in the Abstract should be something like, "At present, school bullying incidents occur frequently, attracting increased attention from researchers."
Response 2: At present, school bullying incidents occur frequently, attracting increased attention from researchers
Point 3 - Delete "by using the questionnaires" after "were surveyed," as this is redundant (line 16). Add a comma and an article in the following: "For controlling any potential confounding factors, a randomized sampling survey method was used to distribute questionnaires." On line 24, "it was observed to be directly associated with students’ perceived school non-physical bullying" (add the word "be"). ETC. –
Response 3: In this study we attempt to explore the impact of parenting styles on perceived school non-physical bullying, 492 students in the fifth and sixth grades of eight primary schools in Zhejiang province were surveyed. For controlling any potential confounding factors, a randomized sampling survey method was used to distribute questionnaires. The results showed that negative affect experiences, negative coping styles, negative family parenting styles, and the perceived school non-physical bullying were all positively correlated with each other (p < 0.05). Perceived verbal bullying differed significantly by sex, grade, and only/non-only children (p < 0.05). Perceived relationship bullying significantly differed between grades (p < 0.05). The sex difference in the perceived cyberbullying also reached a significant level (p < 0.05).The rejection parenting style was shown to be an important factor that may be associated with students’ perceived school non-physical bullying; it was observed to be directly associated with students’ perceived school non-physical bullying and indirectly associated with students’ perceived school non-physical bullying by influencing negative affect experiences and negative coping styles. In conclusion, negative affect experiences and coping styles may have a chain-like mediating effect between the rejection parenting style and students’ perceived school verbal bullying. Moreover, negative affect experiences may have a partial mediating effect between the rejection parenting style and students’ perceived school cyberbullying, relationship bullying, and non-physical bullying total scores. The results of this study provide first-hand empirical data support for schools, families and education authorities to guide and manage non-physical bullying incidents in schools. They also provide a theoretical basis for the subsequent related researches in the field of non-physical bullying.
Introduction -
The Introduction is too long, going into excessive detail for every construct. It should be edited down and include only major references that are the most recent.
Point 1 - As noted in the first review, using the term "victim" is pejorative for targets of bullying. The authors would like to continue to use this term, and state: "The victims here include target of bullying and the victim. For brevity, we will use victim instead of both in our article" on lines 41-42. This doesn't make sense and does not capture the point. Instead, if the authors and editors decide to continue to use the term "victim," they can state something like, ""Victim" will be used throughout this manuscript rather than "target of bullying" for brevity's sake."
Response 1: School bullying is unwelcome offensive behavior, and victims (The victims here include target of bullying and the victim. "Victim" will be used throughout this article rather than "target of bullying" for brevity's sake.") may endure physical, psychological, social, or educational harm [2].
Point 2 - This sentence does not make any sense. Please reword to make the meaning more clear: (lines 64-65) "Studies have shown the link between the Perceived verbal bullying and self-harm behavior: the perceived verbal bullying in non-physical bullying greatly increases the possibility of self-harm among young people."
Response 2: Research shows a link between perceived verbal bullying and self-harm: Perceived verbal bullying in non-physical bullying greatly increases the likelihood of self-harm in young people.
Point 3 - Would recommend not using the phrase "emotionally disturbed" (not appropriate for a study of children) on lines 70-71 - use more precise language, such as experience higher levels of depression and symptoms of anxiety, or whatever was found in the cited study.
Response 3: Perceived relationship bullying can not only make the victim more emotional problems but also lead to some externalizing problems, thus increasing the likelihood of being the target of repeated bullying [28,29].
Point 4: This sentence doesn't make sense - perceived non-physical bullying on school? (page 3, lines 98-99) Please reword - "The family parenting style is an important factor that influences the occurrence of non-physical bullying at school."
Response 4: The family parenting style is an important factor that influences the occurrence of non-physical bullying at school.
Point 5 - "hyperactive-attention deficits" isn't an appropriate term to characterize ADHD - instead, could state "...correlated with hyperactivity and attention deficits " on page 3, lines 107-108)
Response 5: Studies have found that rejection parenting style is significantly positively correlated with children's psychological disorders, conduct problems, hyperactivity and attention deficits; but it is negatively correlated with adolescent prosocial behaviors significantly [36, 37].
Point 6 - Another example of awkward wording: "thus are more likely to perceived non-physical bullying on school" - perhaps the authors mean "thus are more likely to perceive that they are being the target of non-physical bullying at school." (page 3, lines 115-116).
Response 6: Children who experience the rejection parenting style are less likely to establish positive relationships with others outside the family, are more likely to be bullied [39,40], and thus are more likely to perceive that they are being the target of non-physical bullying at school [41].
Methods –
Point 1 - Do not start a sentence with a number (12 students).
Response 1: Twelve students (6 boys and 6 girls) in the fifth and sixth grades of elementary school were randomly selected.
Point 2 - What constituted an "invalid questionnaire"? How many (what percentage) were deemed invalid?
Response 2:
- There are too many missed answers in the questionnaire, generally 2/3 of the total number of missed answers shall prevail.
- The option ticked in the whole questionnaire is the same.
- The options selected in the whole questionnaire are regular, for example, if the options are filled in 1, 2, 3, 1, 2, 3, 1, 2, 3, the questionnaire is invalid.
A total of 12.14% of the questionnaires were considered invalid questionnaires.
Point 3 - Recommend that the authors do not use the word "subjects" on page 6 line 283, but instead use the word "children." "Subjects" is more appropriate to animals in experimental research.
Response 3: 2.3.1. Negative Parenting Styles Scale
The Egna Minnen Beträffende Uppfostran Questionnaire (EMBU) was used to measure negative parenting styles. This questionnaire was originally developed by Perris et al. as one of the commonly used tools to measure parenting styles [62]. The standard version of the EMBU has a total of 81 questions, asking the children to recall the way their father and mother treated themselves when they were growing up, and evaluate them from four levels of "never", "occasionally", "often" and "always". The questionnaire includes four dimensions: Rejection, Emotional Warmth, Overprotection and Favoring Child. After that, Castro et al. revised the Egna Minnen Beträffende Uppfostran for Children (EMBU-C) for primary school children aged 7-12 [63]. The questionnaire has a total of 41 questions, including four dimensions of "rejection", "emotional warmth", "overprotection" and "preferring children ". Due to the good reliability and validity of the EMBU-C and its suitability for younger primary school children, the questionnaire has been widely used and revised worldwide [64,65]. Muris revised EMBU-C, deleted the dimension of "preferring children " and added the dimension of "anxious rearing". The revised 40-item EMBU-C includes four dimensions, "Rejection", "Emotional Warmth", "Anxious Rearing" and "Overprotection", with good reliability and validity, widely used worldwide [66].
Considering the characteristics of the social situation in which students live and vertical development differences. The Egna Minnen Beträffende Uppfostran Questionnaire revised by Wang et al. in 2018 was used in this study [67,68]. The scale asks children to
Point 4 - For the Negative Affect Experience Scale, what is the time period over which the participants are rating experiencing the negative emotions?
Response 4: Since the start of the school (The length of time is matched against the length of time of non-physical bullying)
Discussion -
Point 1 - In my first review, I mentioned not using causative words like "impact," as this is a cross-sectional study. However, the word impact still appears throughout the manuscript, including in the first sentence of the Discussion. Please search for this word and add in something like "association" or "relationships" - i.e., "to explore the relationships of parenting styles on the students' perceived school non-physical bullying..." All of these constructs are being measured at the same time, so causation cannot be implied, correct? It's also not appropriate to say that one variable had a "significant influence" on another (see line 15, line 533). Instead, the variables were related, associated, or correlated with each other.
Response 1: From the perspective of research design and control, this is a cross-sectional questionnaire study, but the questionnaire study can also implement causal elements. By determining the sequence of variables and conducting scientific and appropriate statistical analysis on the data, the causal relationship Inferences can be obtained from statistical control. According to Mill's logic of causal inference (Mill, 2002/1848), after knowing the sequence of variables, the causal relationship between variables can be inferred. The covariation method in Mill's logic allows one effect to have multiple causes, and the extension of causality is very important for psychology and other social science research, because most of the reasons considered in social science and psychological research are only part of the reason, not the only reason.
This questionnaire study can be based on a common event series. Compared with Y, X is the earlier event to determine which variable comes first. From the sequence of events, the rejection parenting style (X) comes first, negative emotional experiences and negative coping styles come second, and the perceived non-physical bullying incident (Y) comes last.
According to previous research and common sense, by excluding the possible confounding effects (Control variables included in this study, such as gender, grade, and whether to be an only child), statistical control was performed through hierarchical regression analysis. After that,the corresponding causal analysis inferences can be made.
In addition, due to the ethical consideration of non-physical bullying research, the limitations of the researcher's current research ability and the research conditions when this study was carried out, it is impossible to demonstrate this concept through experimental research. This is the direction that needs further consideration and improvement in the follow-up research. For example, independent, mediating, and dependent variable questionnaires can be delivered at different time points to try to avoid the limitations of cross-sectional questionnaire research.
In conclusion, the statistical control is more important in questionnaire researches because its variable control is less stringent than experimental researches. In this study, the target of causal inference is achieved by the Mill logic of causal inference and the statistical control.
(1. Mill, J. S. (2002/1848). System of logic: Ratiocinative and inductive. Honolulu, Hawaii: University Press of the Pacific.
- Zhong-Lin Wen. (2017) Causal Reasoning and Analysis in Empirical Research. Journal of Psychological Science ((01), 200-208. doi: 10.16719/j.cnki.1671-6981.20170130.)
Point 2 - The Discussion is also extremely long and would benefit from a more concise presentation to read like a research article.
Response 2: The Discussion section was revised based on reviewer 2's suggestion, with subsequent additions and extensions.
Comments of reviewer 2: Discussion Review Points
- The discussion requires an opening paragraph that describes the study.
- The discussion should be organized by the hypotheses, not by the analyses performed.
- The discussion should integrate the results with past published research and provides a clear explanation of the results, including their importance.
- The discussion does not include: generalizability of the findings, limitations and future research.
References –
Point 1: While the authors believe that including dissertations is justified because there are experts on the committees, there is a major problem with potential bias and conflict of interest. The members of dissertation committees are oftentimes at the same program and institution and some are even in the same lab or research group. The expert committee chair and members have a vested interest in having the person defending the dissertation pass and succeed. I will defer to the Editors, but I would recommend only citing research studies that have been published in scientific journals and have gone through a review of peers that is unbiased and free from conflicts of interest.
Response 1:
There are three stages to pass master’s and doctoral dissertations in China. The first stage is pre-blind review, anonymous review, and double-blind review by experts. A thesis is sent to three on-campus subject experts for review. According to the evaluation results of the three experts, it is determined by comprehensive consideration whether the academic level of the dissertation has reached the level of a master's or doctoral graduate; only then can it be eligible for external review. In the second stage, the off-campus blind review, the evaluation and review are also submitted anonymously, the expert double-blind review, and a thesis is sent for review to three different subject experts in three institutions of higher learning of the same level or higher, and whether the academic level of the dissertation reaches the level of master's or doctoral graduates is determined according to the evaluation results of the three experts. This review rule is designed to avoid potential biases and conflicts of interest. The third stage is the dissertation defense stage, and only this stage has the chairman and members of the expert committee. At this time, the thesis has been perfected almost; and the expert committee cannot be a member of the same laboratory or research group. The defense stage is not only a stage to review whether the dissertation has reached the level of a master's or doctoral dissertation, but also a stage of comprehensive improvement of the dissertation. Such research results should be reliable and credible.
Thank you very much for your guidance and help!
PS: English Editing Certificate is attached.

This manuscript is a resubmission of an earlier submission. The following is a list of the peer review reports and author responses from that submission.
Round 1
Reviewer 1 Report
Title – This title is much too long and very confusing – what does this mean, “Refusal parenting style is easier for students to perceive non-physical bullying?” - instead recommend – “Perceived non-physical bullying in Chinese elementary school students: Parenting style, negative affect and coping as predictors”
Abstract –
Point 1 - As this is a cross-sectional study, it is not appropriate to use causal language such as “impact.” Please make certain to change this throughout – parenting style might be associated w/perception of bullying, but you cannot make the inference that parenting style causes these perceptions.
Point 2 - It is not apparent from the Abstract that any of the analyses were run while controlling for any potential confounders.
Point 3 – The use of the word “campus” to refer to bullying that takes place at school might be somewhat confusing for U.S. readers. This word is used for colleges and universities and not for elementary schools. Consider replacing this or explaining what “campus” means in China.
Introduction
Point 1 – the definition of the main variable of interest – “perception of campus non-physical bullying” is awkwardly worded – “Perception of campus non-physical bullying refers to the process of mental activity in which the victim realizes and believes that they have been subjected to campus non-physical bullying, which can significantly impact the victim.” Instead, consider something like the following: “Perceived non-physical school bullying refers to a student realizing and believing that they have been subjected to non-physical school bullying.”
Point 2 – Generally, the field of bullying research is moving away from characterizing those who are the targets of bullying as “victims.” Consider instead, the preferred terminology of “target” or “target of bullying.” Using the word “victim” implies that the person is weaker and has been harmed by the attempt at being bullied, which is not necessarily accurate. In this manuscript, “victim” is used 37 times.
Point 3 – this is a really confusing paragraph: “According to the results of previous interviews, although some senior elementary school students perceive some campus non-physical bullying phenomena and behaviors, some students do not understand that non-physical bullying is a kind of campus bullying. Those students may be indifferent to non-physical bullying behaviors, such as “joking and satire,” “unpleasant nicknames,” and “isolation and rejection,” or they may think these are jokes among classmates. Some even reported thinking that “posting ugly photos and malicious messages on the Internet” and” depreciating the characteristics of others” are funny. It was seen that there was a huge difference in the perception of campus non-physical bullying among students in the upper elementary school grades.”
Particularly, the phrase…”do not understand that non-physical bullying is a kind of campus bullying” is confusing. If they do not perceive themselves as being the target of bullying, then they are not. They can think that some of those acts are funny. It is not up to the authors of this study to determine what is or is not perceived as bullying by an intended target. Please reword this paragraph to allow for different experiences to be valid.
Point 4 – From what literature are these parenting styles taken from? The “refusal parenting style” is not well known in the child psychology literature. This is a major concern regarding this study. Parenting styles that are more often studied include authoritative, authoritarian, passive, and neglectful (Baumrind). Please provide more references and justify why these parenting styles were chosen rather than others with a greater research literature.
Point 5 – In the Negative Coping Styles section – this sentence is confusing and needs to be rewritten – “Foreign researchers found that emotional problems are positively related to coping styles” – perhaps refer to the country from which the sample(s) were recruited rather than saying “foreign researchers.”
Point 6 – In the last paragraph of the Introduction was the phrase - …”we attempted to explore the mode and action force” – this is not often used in academic articles and is not clear. Please reword.
Methods
Point 1 –Please give more examples of items for the different parenting styles, particularly for the “refusal” parenting style that is not well known and is a major focus of the findings.
Point 2 – In the statistical analysis section, much more detail is needed, including which variables were included as confounders. Also in the Methods how demographics (gender, grade, only child status) were determined from the students should be added as a measure. Did the children report on this? Was it taken from school records?
Results
- Was gender identity or biological sex assessed? If only “sex” information gathered, please change the word “gender” to “sex” throughout.
- It is not appropriate to include dissertations as references in academic published articles as these have not been through peer and editorial review. I would recommend omitting all SIX of these dissertations as references. Also when referring to authors in the text, use the last name only and then put in either “et al.” or “and colleagues,” followed by the year.
- “Refusal” type of parenting might also impact children’s self concept and make them act in ways in which others perceive them differently and make them more likely to be a target of bullying.
Discussion
This section is well written but some of the recommendations seem to expand beyond the findings. Please try to be as conservative as possible and only make direct recommendations that tie into your findings for children, parents, schools, etc.
Review of “Refusal Parenting Style is Easier for Students to Perceive Non-physical Bullying in the Upper Grades of Elementary School: Mediation Effect of Negative Affect Experience and Negative Coping Style”
Title – This title is much too long and very confusing – what does this mean, “Refusal parenting style is easier for students to perceive non-physical bullying?” - instead recommend – “Perceived non-physical bullying in Chinese elementary school students: Parenting style, negative affect and coping as predictors”
Abstract –
Point 1 - As this is a cross-sectional study, it is not appropriate to use causal language such as “impact.” Please make certain to change this throughout – parenting style might be associated w/perception of bullying, but you cannot make the inference that parenting style causes these perceptions.
Point 2 - It is not apparent from the Abstract that any of the analyses were run while controlling for any potential confounders.
Point 3 – The use of the word “campus” to refer to bullying that takes place at school might be somewhat confusing for U.S. readers. This word is used for colleges and universities and not for elementary schools. Consider replacing this or explaining what “campus” means in China.
Introduction
Point 1 – the definition of the main variable of interest – “perception of campus non-physical bullying” is awkwardly worded – “Perception of campus non-physical bullying refers to the process of mental activity in which the victim realizes and believes that they have been subjected to campus non-physical bullying, which can significantly impact the victim.” Instead, consider something like the following: “Perceived non-physical school bullying refers to a student realizing and believing that they have been subjected to non-physical school bullying.”
Point 2 – Generally, the field of bullying research is moving away from characterizing those who are the targets of bullying as “victims.” Consider instead, the preferred terminology of “target” or “target of bullying.” Using the word “victim” implies that the person is weaker and has been harmed by the attempt at being bullied, which is not necessarily accurate. In this manuscript, “victim” is used 37 times.
Point 3 – this is a really confusing paragraph: “According to the results of previous interviews, although some senior elementary school students perceive some campus non-physical bullying phenomena and behaviors, some students do not understand that non-physical bullying is a kind of campus bullying. Those students may be indifferent to non-physical bullying behaviors, such as “joking and satire,” “unpleasant nicknames,” and “isolation and rejection,” or they may think these are jokes among classmates. Some even reported thinking that “posting ugly photos and malicious messages on the Internet” and” depreciating the characteristics of others” are funny. It was seen that there was a huge difference in the perception of campus non-physical bullying among students in the upper elementary school grades.”
Particularly, the phrase…”do not understand that non-physical bullying is a kind of campus bullying” is confusing. If they do not perceive themselves as being the target of bullying, then they are not. They can think that some of those acts are funny. It is not up to the authors of this study to determine what is or is not perceived as bullying by an intended target. Please reword this paragraph to allow for different experiences to be valid.
Point 4 – From what literature are these parenting styles taken from? The “refusal parenting style” is not well known in the child psychology literature. This is a major concern regarding this study. Parenting styles that are more often studied include authoritative, authoritarian, passive, and neglectful (Baumrind). Please provide more references and justify why these parenting styles were chosen rather than others with a greater research literature.
Point 5 – In the Negative Coping Styles section – this sentence is confusing and needs to be rewritten – “Foreign researchers found that emotional problems are positively related to coping styles” – perhaps refer to the country from which the sample(s) were recruited rather than saying “foreign researchers.”
Point 6 – In the last paragraph of the Introduction was the phrase - …”we attempted to explore the mode and action force” – this is not often used in academic articles and is not clear. Please reword.
Methods
Point 1 –Please give more examples of items for the different parenting styles, particularly for the “refusal” parenting style that is not well known and is a major focus of the findings.
Point 2 – In the statistical analysis section, much more detail is needed, including which variables were included as confounders. Also in the Methods how demographics (gender, grade, only child status) were determined from the students should be added as a measure. Did the children report on this? Was it taken from school records?
Results
- Was gender identity or biological sex assessed? If only “sex” information gathered, please change the word “gender” to “sex” throughout.
- It is not appropriate to include dissertations as references in academic published articles as these have not been through peer and editorial review. I would recommend omitting all SIX of these dissertations as references. Also when referring to authors in the text, use the last name only and then put in either “et al.” or “and colleagues,” followed by the year.
- “Refusal” type of parenting might also impact children’s self concept and make them act in ways in which others perceive them differently and make them more likely to be a target of bullying.
Discussion
This section is well written but some of the recommendations seem to expand beyond the findings. Please try to be as conservative as possible and only make direct recommendations that tie into your findings for children, parents, schools, etc.
Author Response
Response to Reviewer 1 Comments
Abstract –
Point 1: As this is a cross-sectional study, it is not appropriate to use causal language such as “impact.” Please make certain to change this throughout–parenting style might be associated w/perception of bullying, but you cannot make the inference that parenting style causes these perceptions.
Response 1: The relevant contents have been revised as follows:
The refusal parenting style was shown to be an important factor that may be associated with students’ perceived school non-physical bullying; it was observed to directly associated with students’ perceived school non-physical bullying and indirectly associated with students’ perceived school non-physical bullying by influencing negative affect experiences and negative coping styles.
Point 2: It is not apparent from the Abstract that any of the analyses were run while controlling for any potential confounders.
Response 2: Relevant contents have been added as follows:
For controlling any potential confounding factors randomized sampling survey method was used to distribute questionnaires.
Other relevant information about potential confounders is in the research design and data analysis section. Please see them.
Point 3: The use of the word” campus” to refer to bullying that takes place at school might be somewhat confusing for U.S. readers. This word is used for colleges and universities and not for elementary schools. Consider replacing this or explaining what “campus” means in China.
Response 3: "School" replace of "campus" has been changed in the full text.
Introduction
Point 1: the definition of the main variable of interest–”perception of campus non-physical bullying” is awkwardly worded–”Perception of campus non-physical bullying refers to the process of mental activity in which the victim realizes and believes that they have been subjected to campus non-physical bullying, which can significantly impact the victim.” Instead, consider something like the following: “Perceived non-physical school bullying refers to a student realizing and believing that they have been subjected to non-physical school bullying.”
Response 1: we adopted your new definition and changed in the article. On p2, paragraph two
Perceived school non-physical bullying refers to a student or a group realizing and believing that they have been subjected to school non-physical bullying. which makes his or their mental health suffer a certain degree of adverse effects.
Point 2: Generally, the field of bullying research is moving away from characterizing those who are the targets of bullying as “victims.” Consider instead, the preferred terminology of “target” or “target of bullying.” Using the word “victim” implies that the person is weaker and has been harmed by the attempt at being bullied, which is not necessarily accurate. In this manuscript, “victim” is used 37 times.
Response 2: We adopted your proposal and changed it to:
School bullying is unwelcome offensive behavior, and victims (The victims here include target of bullying and the victim. For brevity, we will use victim instead of both in the following text) may endure physical, psychological, social, or educational harm .
Point 3: this is a really confusing paragraph: “According to the results of previous interviews, although some senior elementary school students perceive some campus non-physical bullying phenomena and behaviors, some students do not understand that non-physical bullying is a kind of campus bullying. Those students may be indifferent to non-physical bullying behaviors, such as “joking and satire,” “unpleasant nicknames,” and “isolation and rejection,” or they may think these are jokes among classmates. Some even reported thinking that “posting ugly photos and malicious messages on the Internet” and” depreciating the characteristics of others” are funny. It was seen that there was a huge difference in the perception of campus non-physical bullying among students in the upper elementary school grades.”
Particularly, the phrase…”do not understand that non-physical bullying is a kind of campus bullying” is confusing. If they do not perceive themselves as being the target of bullying, then they are not. They can think that some of those acts are funny. It is not up to the authors of this study to determine what is or is not perceived as bullying by an intended target. Please reword this paragraph to allow for different experiences to be valid.
Response 3: To investigate the current situation of non-physical bullying among urban primary school students in China, after the literature study, an interview study was conducted (see the research methods section for research design and basic information of research objects). From the interview research, we found that when faced with the same bullying behaviors, such as “others call me bad nicknames, or make fun of and ridicule me,” different interviewees had different interpretations of these similar or the same behaviors, Some respondents thought they were hurt a lot, some thought it was a funny joke, and some thought it was just normal human interaction. The interviewees' interpretations and responses to non-physical bullying among classmates were quite different among students in the upper grades of elementary school. Whether students perceived non-physical bullying depended not only on the behaviors themselves, but also appeared to be influenced by other factors.
Point 4: From what literature are these parenting styles taken from? The “refusal parenting style” is not well known in the child psychology literature. This is a major concern regarding this study. Parenting styles that are more often studied include authoritative, authoritarian, passive, and neglectful (Baumrind). Please provide more references and justify why these parenting styles were chosen rather than others with a greater research literature.
Response 4:
The family parenting style is an important factor that influences the occurrence of non-physical bullying on school [30–32]. A negative parenting style refers to a kind of stable negative behavior tendency that accompanies parents in the process of educating their children [33], including the three dimensions of refuasl, overprotection, and anxiety. Evidence shows that when children experience positive parenting styles, parents may be more sensitive to their children’s needs, thereby improving the relationships with their children. Improvements in this relationship can cultivate children’s emotional regulation and problem-solving abilities [34,35], as well as reduce negative affect experiences and negative coping styles. Studies have found that refusal parenting style is significantly positively correlated with children's psychological disorders, conduct problems, and hyperactivity-attention deficits, and significantly negatively correlated with adolescent prosocial behaviors[36, 37]. The essence of Carl Rogers' theory is that acceptance and unconditional positive regard is the basis for mental health and that refuasl is the basis for psychological disturbances[38]. Refuasl may not only hurt the self-concept and undermine children’s feelings of relatedness to their parents, but also result in a sense of alienation from the child’s authentic self. Children who experience the refusal parenting style are less likely to establish positive relationships with others outside the family, are more likely to be bullied [39,40], and thus are more likely to perceived non-physical bullying on school[41]. Previous studies have also found that children who are overprotected by their parents may not develop qualities such as autonomy and advocacy, overprotection in the negative parenting style increases the probability that children perceive school bullying, including non-physical bullying [42]. The refusal and anxious negative parenting styles are related to children’s perceived cyberbullying in the context of school non-physical bullying. These excessive or neglected parenting styles comprise one of the variables regarding the perceived cyberbullying [43]. (See the second paragraph of page 2)
2.3.1. Negative Parenting Styles Scale
The Egna Minnen Beträffende Uppfostran Questionnaire (EMBU) was used to measure negative parenting styles. This questionnaire was originally developed by Perris et al. as one of the commonly used tools to measure parenting styles [62]. The standard version of the EMBU has a total of 81 questions, asking the subjects to recall the way their father and mother treated themselves when they were growing up, and evaluate them from four levels of "never", "occasionally", "often" and "always". The questionnaire includes four dimensions: Refuasl, Emotional Warmth, Overprotection and Favoring Subject. After that, Castro et al. revised the Egna Minnen Beträffende Uppfostran for Children (EMBU-C) for primary school children aged 7-12 [63]. The questionnaire has a total of 41 questions, including four dimensions of "refuasl", "emotional warmth", "overprotection" and "preferring subjects". Due to the good reliability and validity of the EMBU-C and its suitability for younger primary school children, the questionnaire has been widely used and revised worldwide [64, 65]. Muris revised EMBU-C, deleted the dimension of "preferring subjects" and added the dimension of "anxious rearing". The revised 40-item EMBU-C includes four dimensions, "Refuasl", "Emotional Warmth", "Anxious Rearing" and "Overprotection", with good reliability and validity, widely used worldwide [66].
(See the last paragraph of page 6)
Point 5: In the Negative Coping Styles section–this sentence is confusing and needs to be rewritten–“Foreign researchers found that emotional problems are positively related to coping styles”–perhaps refer to the country from which the sample(s) were recruited rather than saying “foreign researchers.” Point 6 – In the last paragraph of the Introduction was the phrase - …”we attempted to explore the mode and action force” – this is not often used in academic articles and is not clear. Please reword.
Response 5: The above-mentioned sentence has been changed to:
Some researchers found that emotional problems are positively related to coping styles.
In the last paragraph of the Introduction, we revised it as follows:
Based on the results of the above literature research, we found a chain of causality. There has been little research on the effects of parenting style, negative affect experiences, and negative coping styles on the perceived school non-physical bullying. Therefore, we attempted to explore the associations of parenting style, negative affect experience, and negative coping styles on the perceived school non-physical bullying in senior primary school students, where we hypothesized that negative emotional experience and negative coping styles have mediating effects on the influence of family parenting style on the perceived school non-physical bullying. The study purpose was to draw corresponding conclusions and enrich empirical research in relevant fields through the analysis of the questionnaire scores regarding pupils’ negative coping styles, negative affect experiences, parenting styles, and non-physical bullying scores. This study can provide a reference for the school management and family education intervention of high-grade students’ perceived school non-physical bullying.
Methods
Point 1: Please give more examples of items for the different parenting styles, particularly for the “refusal” parenting style that is not well known and is a major focus of the findings.
Response 1: Considering the characteristics of the social situation in which students live and vertical development differences. The Egna Minnen Beträffende Uppfostran Questionnaire revised by Wang et al. in 2018 was used in this study [67,68]. The scale asks subjects to recall the way their fathers and mothers treated them when they were growing up, as well as to rate them on four levels: “never,” “occasionally,” “often,” and “always.” The questionnaire included three dimensions: refusal parenting style, overprotective parenting style, and anxious parenting style. Scale items include “Dad/Mum often tells me that he doesn't like my behavior at home” “Mom/Daddy treats me unfairly” “Anything I do wrong will be punished by my dad/mum” “I'll be disappointed in dad/mum for not giving me what I want” “I feel like my dad/mom is mean and stingy towards me”, etc. The questionnaire had good reliability, with alpha coefficients ranging from 0.51 to 0.86 for the three dimensions and split-half reliabilities ranging from 0.57 to 0.89. (See the first paragraph of page 7)
Point 2: In the statistical analysis section, much more detail is needed, including which variables were included as confounders. Also, in the Methods how demographics (gender, grade, only child status) were determined from the students should be added as a measure. Did the children report on this? Was it taken from school records?
Response 2:
In this study, sex, grade and only child status is the confounding variable.
Dr. Houyu Zhou have many years of experience in primary education and teaching guidance. Every year, at least half a year, Dr. Houyu Zhou communicated and talked with pupils and teachers of all grades in primary schools and observe the children’s behaviors. In long-term elementary school work practice, she observed and learned these features of the students, such as, sex, grade and only child status, may be the confounding factor on perceived school non-physical bullying of students. Then further based on field observation, experience and literature, we propose these confounding variables in this research design.
2.1.2 Questionnaire research stage
In the questionnaire research phase, the survey was administered in class units. The informed consent form for filling out the questionnaire would be sent to the parents of the students one day in advance by the class teacher. After the parents confirmed their consent, the students were asked to anonymously fill in the questionnaires ensuring that they understood the requirements. The filling-in time was 45 min in total, and all questionnaires were collected on the spot after the respondents had completed them. Following collection, the invalid questionnaires were eliminated and valid questionnaires were collated. This study was approved by the Academic Ethics Committee of Jing Hengyi Institute of Education of Hangzhou Normal University (Approval No: 2021002).
(See the second paragraph of page 6)
Results
Point 1:
- Was gender identity or biological sex assessed? If only “sex” information gathered, please change the word “gender” to “sex”
Response 1: The relevant contents have been revised according to your advice
Such as Tables 1,2,4, etc., as well as relevant parts in the article.
- It is not appropriate to include dissertations as references in academic published articles as these have not been through peer and editorial review. I would recommend omitting all SIX of these dissertations as references. Also when referring to authors in the text, use the last name only and then put in either “et al.” or “and colleagues,” followed by the year.
Response 2:
In China, dissertations are reviewed by at least 3 peer experts before the defense, and at least 4-5 experts participate in the review and defense, so we keep dissertations as references. Thank you very much for your reminding and help!
The relevant contents have been revised according to your advice.
Wójcik & Rzeńca(2021)found Significant link between negative coping and being bullied[61]. Other research reported that bullying victim recognition was significantly correlated with the degree of parent-child relationship (Zhao et al., 2021) [ 72]. Rauschenberg et al. (2020) found bullied people reported extremely strong negative emotional experiences [73]. These findings confirmed hypotheses 1 ,2 and 4. (The related content has been adjusted to the discussion section.)
- “Refusal” type of parenting might also impact children’s self-concept and make them act in ways in which others perceive them differently and make them more likely to be a target of bullying
Response 3: Yes, we quite agree with this view. We add it to the future research direction
Thirdly, refusal” type of parenting might also impact children’s self-concept and make them act in ways in which others perceive them differently and make them more likely to be a target of bullying. This is also the direction that future research needs to further verify.
Discussion
This section is well written but some of the recommendations seem to expand beyond the findings. Please try to be as conservative as possible and only make direct recommendations that tie into your findings for children, parents, schools, etc.
Response 1: This part has been readjusted and rewritten
- Discussion
In this study, we aimed to explore the impact of parenting styles on the students’ perceived school non-physical bullying, and the effects of negative affect experience and negative coping style between them. Correlation analysis shows that negative affect experience, negative coping style, negative parenting style and the perceived school non-physical bullying are all positively related with each other, These findings are consistent with those of recent studies [61, 72, 73]. Wójcik & Rzeńca(2021)found Significant link between negative coping and being bullied[61]. Other research reported that bullying victim recognition was significantly correlated with the degree of parent-child relationship (Zhao et al., 2021) [ 72]. Rauschenberg et al. (2020) found bullied people reported extremely strong negative emotional experiences [73]. These findings confirmed hypotheses 1 ,2 and 4.
The results of multiple hierarchical regression analysis showed that, after controlling the confounding variables such as sex, grade, and only child or not, refusa1 parenting style, negative emotional experience and negative coping style still had significant influence on students' perceived school non-physical bullying.
The results of mediating effect analysis by structural equation modeling showed that negative affect experiences and negative coping styles had a chain-like mediating effect between the refusal parenting style and students’ perceived school verbal bullying. Th findings confirmed hypotheses 6. Moreover, negative affect experience had a partial mediating effect between the refusal parenting style and students’ perceived school non-physical bullying total score , relationship bullying and cyberbullying [72, 74]. These findings partially confirmed hypothesis 3,7and 8. In this study only hypothesis 5 has not been confirmed.
4.1. Correlation and Regression Analysis of Parenting Styles and the perceived School Non-Physical Bullying
The results of the present study demonstrated significant correlations between negative parenting styles, negative affect experience, negative coping styles, and the perceived school non-physical bullying in senior primary school students. We found significant positive correlations between the anxiety, refusal, overprotective parenting styles; negative emotional experience; and negative coping styles. This was consistent with the research finding of Wu & Liu(2009) showing that fathers’ refusal and overprotective parenting styles may cause students to adopt negative coping styles to deal with problems [75]. Negative affect experience and negative coping styles were shown to be significantly positively correlated with students’ perceived verbal bullying, relationship bullying, and cyberbullying. A similar conclusion was reached by researcher Song(2018), who reported that psychological bullying (including verbal bullying, relationship bullying, and cyberbullying) perceived by students is positively correlated with negative coping styles [76]. Among the three forms of perceiving school non-physical bullying, only the perceived cyberbullying showed an insignificant correlation with anxious parenting styles. Perceived verbal bullying and relationship bullying were shown to be significantly positively correlated with the overprotective, refusal, and anxious parenting styles. In related studies, parents’ overprotective and mothers’ refusal parenting styles were found to have significantly affected junior high school students’ perceived relationship and verbal bullying in the context of school non-physical bullying [77]. Other studies pointed out that parents’ overprotective and refusal parenting styles are significantly positively correlated with junior high school students’ perceived cyberbullying [78]. These research conclusions were consistent with the results of the present study, indicating that the psychological development characteristics of senior primary school students in this regard are at the same levels as those of junior middle school students.
According to the results of multiple hierarchical regression analysis, refusal parenting, negative affect experience, and negative coping styles jointly explained 22.5% of the total variation of perceived verbal bullying (F = 20.637, p < 0.01); the refusal parenting style and negative affect experience jointly explained 19.2% of the total variation of perceived social relationship bullying (F = 15.432, p < 0.01); refusal parenting style and negative affect experience jointly explained 10.2% of the total variation of perceived cyberbullying (F = 8.082, p < 0.01); and negative affect experience and the refusal parenting style jointly explained 21.1% of the total variation of the total score of perceived non-physical bullying (F = 18. 162, p < 0.01). These results demonstrate that the refusal parenting style was an important factor that affected the students’ perceived school non-physical bullying. Parents’ blind criticism or refusal, such as “unreasonable reprimand and punishment” and “unfair and petty treatment,” make children more likely to perceive school non-physical bullying in the school environment. One possible explanation is that parents’ blind refusal and denial in the family environment comprise a situation that is similar to the non-physical verbal, relationship, and cyberbullying experienced by students on school. The mentioned behaviors of parents harm students’ self-esteem and self-confidence, thus leading to self-recognition difficulties and the acceptance of non-physical bullying by others [79]. Negative emotions, such as derogation and unfair treatment previously experienced by students in the family environment, can transfer to experiences similar to school non-physical bullying, and the negative coping styles adopted under the refusal parenting style can transfer to similar coping styles of school non-physical bullying. As a result, students who grow up under the refusal parenting style are more likely to perceive school non-physical bullying.
4.2. The Mediating Effect of Negative Affect Experience and Negative Coping Styles between the Refusal Parenting Style and the Perceived School Non-Physical Bullying
The results of the mediating effect test model showed that negative emotional experience and negative coping styles had a significant chain mediating effect between the refusal parenting style and the students’ perceptions of school verbal bullying, and negative emotional experience significantly positively predicted the perceived school verbal bullying by influencing the negative coping style. These results showed that the refusal parenting style not only directly affected the students’ perceived school verbal bullying but also indirectly affected the students’ perceived school verbal bullying by influencing negative emotional experiences and negative coping styles.
During the process of a student’s growth, parents frequently refuse, deny, and adopt negative attitudes toward them in all aspects of study and life, e.g., “disappointment,” “unfairness,” “pettiness,” and “punished for no reason.” As such, students are placed in a negative family atmosphere and growth environment for a long time, which makes them cold and indifferent, thus generating the perception that they are not cared for by their families. Over time, students grow to feel insecure, thus intensifying their negative affect experiences [80]. When facing problems, they are prone to have negative affect experiences, such as “tension” and “irritability,” which may lead them to adopt simple and negative coping styles when solving problems [81, 82]. This kind of refusal parenting style had negative impacts on students’ affect experiences and coping styles. Our analysis showed that the degrees of negative affect experience and negative coping styles were correlated with the students’ perceptions of school non-physical bullying, and the refusal parenting style was observed to affect students’ perceived school non-physical bullying by influencing students’ negative affect experiences and negative coping style.
The partial mediating effect of negative affect experience between the refusal parenting style and the total score of students’ perceptions of school non-physical bullying, relationship bullying, and cyberbullying suggested that the refusal parenting style not only directly affected students’ perceived school relationship bullying and cyberbullying but also indirectly impacted students’ perceived school relationship bullying and cyberbullying by influencing their negative affect experiences.
The results of the above-mentioned mediating effect analyses indicate that there were differences in the psychological mechanisms of action of the refusal parenting style on students’ perceived school verbal bullying, relationship bullying and cyberbullying.
4.3. Theoretical Implications
In previous bullying studies, the association between Refusal parenting style and perceived school non-physical bullying as well as the mediation effects of negative affect experiences and cope in this association is limited. This research highlights the significant influence of refusal parenting style on students' perceived school non-physical bullying behavior. The nature of the emotional-behavioral bond between parent and child has an important impact on children's interpretation of the behavior of others. This suggests that children in the context of refusal parenting are more likely to establish a link between others’ non-physical bullying behaviors and self-perceived non-physical bullying.
In addition, this study reveals the mediating role of negative affective experiences and coping in students' perceived school non-physical bullying in the context of family refusal parenting. The mediating effect of negative affective experiences and coping shows that refusal parenting style affects negative affect experience, and negative affect experience also easily leads to negative coping style, and the effect of refusal parenting style on perceived non-physical bulling may depend on the level of negative affect experience and coping, especially in students perceived of verbal bullying.
This study enriches the related research results of non-physical bullying in the past from the perspective of refusing parenting style, negative affect experiences and coping. It also provides a theoretical basis and empirical data support for the subsequent related research in the field of non-physical bullying.
- Practical Implications
5.1. Schools
The outcomes of this research suggest some practical implications for the schools. Non-physical bullying mainly occurs in schools, therefore, corresponding measures taken by the school play important roles in students’ perceived school non-physical bullying.
First of all, schools should use parents’ meetings to guide students’ parents to realize the impact of family education on students’ perceived school non-physical bullying, and they should advise parents to rarely, if ever, use the refusal parenting style at home. Second, after students perceive a school non-physical bullying incident, the school should take corresponding measures to prevent the bullying from continuing and carry out effective interventions. Minimize bullied students' negative emotional experience. Such as to pay full attention to the students who are aware of school non-physical bullying, to give the bullied timely and enough guidance and help, and to provide timely and effective psychological counseling, support, and comfort etc.
5.2. Parents
In this study, it was found that the refusal parenting style is an important factor that affected students’ perceptions of school non-physical bullying because students’ negative affect experiences under the refusal parenting style were easily transferred to experiences that were similar to school non-physical bullying. The negative coping styles adopted by students under the refusal parenting style also easily migrated to coping styles similar to school verbal bullying. Therefore, students growing up under the refusal parenting style were more likely to perceive school non-physical bullying. Accordingly, parents should try their best to be cautious with or avoid using the refusal parenting type in the process of educating their children. In addition, to help children reduce the adverse effects of non-physical bullying, parents should also do the following:
First of all, parents should help their children cope with negative emotional experiences. In daily family education parents should pay attention to their children’s emotional expression, identify their abnormal emotions in time, and take appropriate and effective measures to provide their children sufficient companionship, security, love, and care to prevent children from having overly negative school and peer emotional experiences. It is also important for parents to guide children to learn releasing negative emotions and restoring calm and positive emotional experience. If the child's emotional problems are serious enough, parents should take their children to seek the help of psychological professionals, schools, and teachers to help children vent extreme negative emotions in time and avoid extreme behavior.
Secondly, parents should teach their children positive coping styles. For example, they should help children calmly face non-physical bullying, have full confidence, build sufficient and effective coping strategies to resolve difficulties. when students observe the school non-physical bullying of another student, they should be able to provide appropriate and timely help, e.g., stop it or notify teachers, so that relevant non-physical bullying events on school can be actively and properly handled.
Thank you very much for your guidance and help!

Reviewer 2 Report
General Comment Points
This manuscript reports on an interesting topic. I applaud the aspirations represented in this paper. However, both formal and content aspects of the manuscript must be revised. I hope the suggestions I give below will support you in advancing your research efforts on this topic. Following are my specific comments on this paper.
Title Review Points
The title does not captures the reader’s attention and it is too long.
Abstract Review Points
- The authors should include more information on the importance of the topic, the gap and hypotheses or questions under investigation.
- Morevover, the authors should improve the conclusions, implications and applications of the findings.
Introduction Review Points
- The introduction does not present the importance of the problem. It does not include the theoretical and practical implications.
- The introduction should improve the knowledge on the topic of this paper and the reasons for the study presented in this report.
- The end of the introduction presents the aim of the study but it should be reviewed and improved. There are no hypotheses.
Method Review Points
- The description of the participants is too brief.
- The sampling method is not described.
- The description of the instruments does not include examples of items.
- In addition, the authors report that there are two phases to the study. In the first phase they have conducted interviews and included the questions. However, nowhere is the phases of the study explained. Nor is the interview design procedure described.
- With respect to the instruments, because they are implemented on children and adolescents, the informed consent of the parents and children/adolescents should be obtained. The study must also have been approved by an ethics committee. The authors should report on these issues.
- The authors should significantly improve the statistical analysis section.
Discussion Review Points
- The discussion requires an opening paragraph that describes the study.
- The discussion should be organized by the hypotheses, not by the analyses performed.
- The discussion should integrates the results with past published research and provides a clear explanation of the results, including their importance.
- The discussion does not include: generalizability of the findings, limitations and future research.
References Review Points
- Authors should include more recent references
Author Response
Response to Reviewer 2 Comments
Abstract –
Point 1: The authors should include more information on the importance of the topic, the gap and hypotheses or questions under investigation.
Response 1: At present, school bullying incidents occur frequently, which attracts increasing attention of researchers.
Point 2: Morevover, the authors should improve the conclusions, implications and applications of the findings.
Response 2: The results of this study provide first-hand empirical data support for schools, families and education authorities to guide and manage non-physical bullying incidents on schools. It also provides a theoretical basis for the subsequent related research in the field of non-physical bullying.
Introduction
Point 1: The introduction does not present the importance of the problem. It does not include the theoretical and practical implications.
Response 1: The study purpose was to draw corresponding conclusions and enrich empirical research in relevant fields through the analysis of the questionnaire scores regarding pupils’ negative coping styles, negative affect experiences, parenting styles, and non-physical bullying scores. This study can provide a reference for the school management and family education intervention of high-grade students’ perceived school non-physical bullying.
Point 2: The introduction should improve the knowledge on the topic of this paper and the reasons for the study presented in this report.
Response 2: To investigate the current situation of non-physical bullying among urban primary school students in China, after the literature study, an interview study was conducted (see the research methods section for research design and basic information of research objects). From the interview research, we found that when faced with the same bullying behaviors, such as “others call me bad nicknames, or make fun of and ridicule me,” different interviewees had different interpretations of these similar or the same behaviors, Some respondents thought they were hurt a lot, some thought it was a funny joke, and some thought it was just normal human interaction. The interviewees' interpretations and responses to non-physical bullying among classmates were quite different among students in the upper grades of elementary school. Whether students perceived non-physical bullying depended not only on the behaviors themselves, but also appeared to be influenced by other factors.
Based on the results of interviews and previous literature research, this study attempts to explore the reasons for these differences. By exploring the causes and clues of perceived non-physical bullying behaviors, we can improve various behavioral factors that are easy to induce such phenomena in family upbringing and school education and teaching, so as to effectively prevent the causes of poor psychological development of students. The results of this study can enrich the research results of primary school students' non-physical bullying from the perspective of family parenting style, provide empirical data support for the follow-up theoretical research in related fields, and also give specific operational ideas and basis for families and schools to effectively intervene in non-physical bullying. For achieving the above objectives, we explored the mental mechanism of the perceived school non-physical bullying in terms of the following: parenting styles, negative affect experiences, and negative coping styles.
Point 3: The end of the introduction presents the aim of the study but it should be reviewed and improved. There are no hypotheses.
Response 3: Therefore, this study proposes:
Figure 1. Hypothesized model.
Hypothesis 1: Negative parenting styles is positively related to perceived school non-physical bullying.
Hypothesis 2: Negative parenting styles is positively related to negative affect experiences;
Hypothesis 3: Negative parenting styles can affect perceived school non-physical bullying by influencing negative affect experiences.
Hypothesis 4: Negative affect experiences is positively related to negative coping styles.
Hypothesis 5: Negative coping styles have a mediating effect between negative parenting style and students’ perceived school non-physical bullying.
Hypothesis 6: Negative affect experiences and negative coping styles have a mediating effect between negative parenting style and students’ perceived school verbal bullying.
Hypothesis 7: Negative affect experiences and negative coping styles have a mediating effect between negative parenting style and students’ perceived school relationship bullying.
Hypothesis 8: Negative affect experiences and negative coping styles have a mediating effect between negative parenting style and students’ perceived school cyberbullying bullying.
Therefore, the hypothesis model of this study is shown in Figure1.
Methods
Point 1: The description of the participants is too brief. The sampling method is not described.
Response 1:
2.1.2 Questionnaire research stage
In the questionnaire research phase, the survey was administered in class units.The informed consent form for filling out the questionnaire would be sent to the parents of the students one day in advance by the class teacher. After the parents confirmed their consent, the students were asked to anonymously fill in the questionnaires ensuring that they understood the requirements. The filling-in time was 45 min in total, and all questionnaires were collected on the spot after the respondents had completed them. Following collection, the invalid questionnaires were eliminated and valid questionnaires were collated. This study was approved by the Academic Ethics Committee of Jing Hengyi Institute of Education of Hangzhou Normal University (Approval No: 2021002).
Research data were collected from 8 primary school in Zhejiang Province in 2021. The aims of the study were introduced to all respondents at the start guidelines of the questionnaire. The respondents had been told that their provided information will not be revealed to anyone and will solely be used for research purposes by the ethical rules of research. Convenience random sampling method was used to control possible potential confounding factors. Anticipating a relatively lower response rate, a total of 560 questionnaires were distributed, and 492 questionnaires were received. Further analysis considered a total of 492 respondents, with an overall response rate of 87.86%. Details of the demographic data are shown in Table 1.
2..2 Sampling and Data Collection
Table 1. Sample characteristics.
|
Measure |
Items |
Frequency (n) |
Percentage (%) |
|
Sex |
Male |
247 |
50.20 |
|
Female |
245 |
49.80 |
|
|
Grade |
Fifth |
222 |
45.12 |
|
Sixth |
270 |
54.88 |
|
|
only/non-only children |
Yes |
174 |
35.37 |
|
No |
318 |
64.63 |
Point 2: The description of the instruments does not include examples of items.
Response 2:
2.3. Measures
All instruments were psychometrically sound, as evidenced by the sufficient reliabilities of the scales in the current study.
2.3.1. Negative Parenting Styles Scale
The Egna Minnen Beträffende Uppfostran Questionnaire (EMBU) was used to measure negative parenting styles. This questionnaire was originally developed by Perris et al. as one of the commonly used tools to measure parenting styles [62]. The standard version of the EMBU has a total of 81 questions, asking the subjects to recall the way their father and mother treated themselves when they were growing up, and evaluate them from four levels of "never", "occasionally", "often" and "always". The questionnaire includes four dimensions: Refuasl, Emotional Warmth, Overprotection and Favoring Subject. After that, Castro et al. revised the Egna Minnen Beträffende Uppfostran for Children (EMBU-C) for primary school children aged 7-12 [63]. The questionnaire has a total of 41 questions, including four dimensions of "refuasl", "emotional warmth", "overprotection" and "preferring subjects". Due to the good reliability and validity of the EMBU-C and its suitability for younger primary school children, the questionnaire has been widely used and revised worldwide [64, 65]. Muris revised EMBU-C, deleted the dimension of "preferring subjects" and added the dimension of "anxious rearing". The revised 40-item EMBU-C includes four dimensions, "Refuasl", "Emotional Warmth", "Anxious Rearing" and "Overprotection", with good reliability and validity, widely used worldwide [66].
Considering the characteristics of the social situation in which students live and vertical development differences. The Egna Minnen Beträffende Uppfostran Questionnaire revised by Wang et al. in 2018 was used in this study [67,68]. The scale asks subjects to recall the way their fathers and mothers treated them when they were growing up, as well as to rate them on four levels: “never,” “occasionally,” “often,” and “always.” The questionnaire included three dimensions: refusal parenting style, overprotective parenting style, and anxious parenting style. Scale items include “Dad/Mum often tells me that he doesn't like my behavior at home” “Mom/Daddy treats me unfairly” “Anything I do wrong will be punished by my dad/mum” “I'll be disappointed in dad/mum for not giving me what I want” “I feel like my dad/mom is mean and stingy towards me”, etc. The questionnaire had good reliability, with alpha coefficients ranging from 0.51 to 0.86 for the three dimensions and split-half reliabilities ranging from 0.57 to 0.89.
2.3.2. Negative Affect Experience Scale
The Negative Affect Experience Scale, prepared by Watson and revised by Zheng and Wang [46,69], contains a total of nine items. Items are rated with 5-point optional responses ranging from none or very slight (1) to very strong (5). Scale items include “scared” “ashamed” “angry”, etc. The scale has sufficient reliability of 0.84.
2.3.3. Negative Coping Style Scale
The Negative Coping Style Scale of the Xie Brief Coping Style Scale, which consists of 8 items, was used [70]. It has 4 options on a rating scale ranging from “don’t use” (0) to “often use” (3). Scale items include “trying to forget the whole thing” “Fantasy that some kind of miracle might happen to change the status quo” “Attempt to rest or take physical leave to temporarily put the problem (annoyance) aside”, etc. The internal consistency of the scale was satisfactory (α = 0.78).
2.3.4. Delaware Bullying Victimization Scale–Student
A portion of the Delaware Bullying Victimization Scale–Student (DBVS-S) was selected as the instrument [71], with 12 items divided into three dimensions: perceived verbal bullying (4 items), perceived relational bullying (4 items), and perceived cyberbullying (4 items). It had 6 options on the rating scale ranging from never (0) to every day (5). Scale items include “Others call me bad nicknames, or make fun of and ridicule me” “Some classmates posted my ugly photos online” “Classmates told or urged others not to be friends with me”, etc. The internal consistency of the scale was satisfactory (α = 0.839).
Point 3: In addition, the authors report that there are two phases to the study. In the first phase they have conducted interviews and included the questions. However, nowhere is the phases of the study explained. Nor is the interview design procedure described.
Response 3:
- Method
2.1. Research Design and Methods
2.1.1 Interview research stage
During the interview study phase, 12 students (6 boys and 6 girls) from the fifth and sixth elemen-tary school grades, were randomly selected to be inter-viewed about non-physical bullying in school. Among these students, there are four students with excellent grades, four students with average grades and four students with below average grades. The interview questions included: Do any students in the class give nicknames to others? Do the students who are given nicknames approve of this behavior? Do you approve of this behavior? If this happened to you, would you feel bullied? Is there a student in your class that is isolated by other students? Do you know the reason why they were isolated? Are there any students in your class who post pictures of others who were scandalized in the WeChat group? How would you feel if these situations happened to you? These questions were used to examine the basic information of non-physical bullying-related incidents and behaviors that occurred in the up-per elementary school years, as well as to assess the extent to which the interview results fit with the content of the research questionnaire used in this study. The presentation of the questionnaire was fine-tuned based on the interview results to suit the comprehension level of upper elementary school students [51]. Through the interviews, we found that there were big differences in the reported degrees of bullying and awareness of non-physical bullying events [61].
2.1.2 Questionnaire research stage
In the questionnaire research phase, the survey was administered in class units.The informed consent form for filling out the questionnaire would be sent to the parents of the students one day in advance by the class teacher. After the parents confirmed their consent, the students were asked to anonymously fill in the questionnaires ensuring that they understood the requirements. The filling-in time was 45 min in total, and all questionnaires were collected on the spot after the respondents had completed them. Following collection, the invalid questionnaires were eliminated and valid questionnaires were collated. This study was approved by the Academic Ethics Committee of Jing Hengyi Institute of Education of Hangzhou Normal University (Approval No: 2021002).
Research data were collected from 8 primary school in Zhejiang Province in 2021. The aims of the study were introduced to all respondents at the start guidelines of the questionnaire. The respondents had been told that their provided information will not be revealed to anyone and will solely be used for research purposes by the ethical rules of research. Convenience random sampling method was used to control possible potential confounding factors. Anticipating a relatively lower response rate, a total of 560 questionnaires were distributed, and 492 questionnaires were received. Further analysis considered a total of 492 respondents, with an overall response rate of 87.86%. Details of the demographic data are shown in Table 1.
Point 4: With respect to the instruments, because they are implemented on children and adolescents, the informed consent of the parents and children/adolescents should be obtained. The study must also have been approved by an ethics committee. The authors should report on these issues.
Response 4: These data were collected from a primary school in Zhejiang Province. This research was conducted in 2021, and the aims of the study were introduced to all respondents at the start of the questionnaire in the guidelines drafted; moreover, according to the ethical rules of research, respondents had been told that their provided information will not be revealed to anyone and will solely be used for research purposes. The respondents were chosen using a convenience sampling method. Anticipating a relatively lower response rate a total of 560 questionnaires were distributed, and 492 questionnaires were received. A total of 492 respondents were considered from further analysis, and the overall response rate was 87.86%. The detail of the demographics in this study is presented in Table 1.
Point 5: The authors should significantly improve the statistical analysis section.
Response 5:
3.3. Multiple Hierarchical Regression Analysis
First, multiple hierarchical regression analysis was performed using the SPSS-26 to examine the directional dependence of the variables and their respective contribution to explaining dependent variables. Table 4-1 shows the results of the multiple hierarchical regression analysis. First, hierarchical regression analysis was performed with verbal bullying as the dependent variable. In the first step, the control variable is introduced into the regression equation to construct model 1; in the second step, the negative parenting style is also introduced into the regression equation to construct model 2; in the third step, negative affect experience and negative coping style are introduced into the equation, and model 3 is constructed. Model 1 indicated that 3% of the variance of the perceived verbal bullying could be attributed to sex and only child (F = 5.086, p < 0.01). Among them, sex has a significant negative predictive effect on verbal bullying(β=-0.757, p<0.05), and only child has a significant positive predictive effect on verbal bullying(β=1.085, p<0.01). Model 2 indicated that 9.2% of the variance of the perceived verbal bullying could be attributed to refusal parenting style (F = 11.177, p < 0.001). And refusal parenting style has a significant positive predictive effect on verbal bullying (β=0.143, p<0.001). Model 3 indicated that 13.3% of the variance of the perceived verbal bullying could be attributed to negative affect experience and negative coping style (F = 20.637, p < 0.001). Negative affect experience can significantly and positively predict verbal bullying (β=0.227, p<0.001). Negative coping style can significantly and positively predict verbal bullying(β=0.097, p<0.05)
Table 4-1. Multiple hierarchical regression analysis of the perceived non-physical bullying on refusal parenting style, negative affect experience, and negative coping styles (N = 492).
|
Dependent Variable |
Independent Variable |
β |
R2 |
Adjust R2 |
F |
|
Verbal bullying |
Model 1 |
|
0.030 |
0.024 |
5.086** |
|
Sex |
-0.757* |
|
|
|
|
|
Grade |
-0.207 |
|
|
|
|
|
Only child |
1.085** |
|
|
|
|
|
Model 2 |
|
0.122 |
0.111 |
11.177*** |
|
|
Sex |
-0.577 |
|
|
|
|
|
Grade |
-0.121 |
|
|
|
|
|
Only child |
0.735* |
|
|
|
|
|
Anxious parenting style |
-0.039 |
|
|
|
|
|
Refusal parenting style |
0.143*** |
|
|
|
|
|
Over-protection |
0.045 |
|
|
|
|
|
Model 3 |
|
0.255 |
0.243 |
20.637*** |
|
|
Sex |
-0.703* |
|
|
|
|
|
Grade |
-0.090 |
|
|
|
|
|
Only child |
0.270 |
|
|
|
|
|
Anxious parenting style |
-0.046 |
|
|
|
|
|
Refusal parenting style |
0.076** |
|
|
|
|
|
Over-protection |
0.019 |
|
|
|
|
|
Negative affect experience |
0.227*** |
|
|
|
|
|
Negative coping style |
0.097* |
|
|
|
|
|
Cyberbullying |
Model 1 |
|
0.016 |
0.010 |
2.646* |
|
Sex |
-0.570* |
|
|
|
|
|
Grade |
0.192 |
|
|
|
|
|
Only child |
0.324 |
|
|
|
|
|
Model 2 |
|
0.057 |
0.045 |
4.874*** |
|
|
Sex |
-0.496* |
|
|
|
|
|
Grade |
0.232 |
|
|
|
|
|
Only child |
0.161 |
|
|
|
|
|
Anxious parenting style |
-0.020 |
|
|
|
|
|
Refusal parenting style |
0.068*** |
|
|
|
|
|
Over-protection |
0.014 |
|
|
|
|
|
Model 3 |
|
0.118 |
0.104 |
8.082*** |
|
|
Sex |
-0.531* |
|
|
|
|
|
Grade |
0.305 |
|
|
|
|
|
Only child |
-0.069 |
|
|
|
|
|
Anxious parenting style |
-0.023 |
|
|
|
|
|
Refusal parenting style |
0.041* |
|
|
|
|
|
Over-protection |
0.004 |
|
|
|
|
|
Negative affect experience |
0.123*** |
|
|
|
|
|
Negative coping style |
-0.026 |
|
|
|
* p < 0.05, ** p < 0.01, *** p < 0.001.
Second, multiple hierarchical regression analysis was performed with cyberbullying as the dependent variable. Table 4-1 shows the results. In the first step, the control variable is introduced into the regression equation to construct model 1; in the second step, the negative parenting style is also introduced into the regression equation to construct model 2; in the third step, negative affect experience and negative coping style are introduced into the equation, and model 3 is constructed. Model 1 indicated that 1.6% of the variance of the perceived cyberbullying could be attributed to sex (F = 2.646, p < 0.05). Sex has a significant negative predictive effect on cyberbullying (β=-0.570, p<0.05). Model 2 indicated that 4.1% of the variance of the perceived cyberbullying could be attributed to refusal parenting style of the negative parenting style (F = 4.874, p<0.001). Refusal parenting style has a significant positive predictive effect on cyberbullying (β=0.068, p<0.001). Model 3 indicated that 6.1% of the variance of the perceived cyberbullying could be attributed to negative affect experience (F = 8.082, p < 0.001). Negative affect experience can significantly and positively predict cyberbullying (β=0.123, p<0.001).
Third, multiple hierarchical regression analysis was performed with relationship bullying as the dependent variable. Table 4-2 shows the results. In the first step, the control variable is introduced into the regression equation to construct model 1; in the second step, the negative parenting style is also introduced into the regression equation to construct model 2; in the third step, negative affect experience and negative coping style are introduced into the equation, and model 3 is constructed. Model 1 indicated that 1.2% of the variance of the perceived relationship bullying could be attributed to sex、grade and only child (F = 1.950, p>0.05), but the regression coefficients of all predicted variables are not significant. The regression equation is not valid. Model 2 indicated that 10.5% of the variance of the perceived relationship bullying could be attributed to refusal parenting style of the negative parenting style (F = 10.642, p<0.001). Refusal parenting style has a significant positive predictive effect on cyberbullying (β=0.123, p<0.001). Model 3 indicated that 8.7% of the variance of the perceived relationship bullying could be attributed to negative affect experience (F = 15.432, p < 0.001). Negative affect experience can significantly and positively predict cyberbullying (β=0.167, p<0.001).
Table 4-2. Multiple hierarchical regression analysis of the perceived non-physical bullying on negative parenting style, negative affect experience, and negative coping styles (N = 492).
|
Dependent Variable |
Independent Variable |
β |
R2 |
Adjust R2 |
F |
|
Relationship bullying |
Model 1 |
|
0.012 |
0.006 |
1.950 |
|
Sex |
-0.427 |
|
|
|
|
|
Grade |
-0.299 |
|
|
|
|
|
Only child |
0.371 |
|
|
|
|
|
Model 2 |
|
0.117 |
0.106 |
10.642*** |
|
|
Sex |
-0.264 |
|
|
|
|
|
Grade |
-0.225 |
|
|
|
|
|
Only child |
0.077 |
|
|
|
|
|
Anxious parenting style |
-0.039 |
|
|
|
|
|
Refusal parenting style |
0.123*** |
|
|
|
|
|
Over-protection |
0.049 |
|
|
|
|
|
Model 3 |
|
0.204 |
0.191 |
15.432*** |
|
|
Sex |
-0.335 |
|
|
|
|
|
Grade |
-0.165 |
|
|
|
|
|
Only child |
-0.250 |
|
|
|
|
|
Anxious parenting style |
-0.043 |
|
|
|
|
|
Refusal parenting style |
0.079*** |
|
|
|
|
|
Over-protection |
0.032 |
|
|
|
|
|
Negative affect experience |
0.167*** |
|
|
|
|
|
Negative coping style |
0.020 |
|
|
|
|
|
Non-physical bullying total score |
Model 1 |
|
0.021 |
0.015 |
3.448* |
|
Sex |
-0.146* |
|
|
|
|
|
Grade |
-0.026 |
|
|
|
|
|
Only child |
0.148* |
|
|
|
|
|
Model 2 |
|
0.117 |
0.106 |
10.719*** |
|
|
Sex |
-0.111 |
|
|
|
|
|
Grade |
-0.010 |
|
|
|
|
|
Only child |
0.081 |
|
|
|
|
|
Anxious parenting style |
-0.008 |
|
|
|
|
|
Refusal parenting style |
0.028*** |
|
|
|
|
|
Over-protection |
0.009 |
|
|
|
|
|
Model 3 |
|
0.232 |
0.219 |
18.162*** |
|
|
Sex |
-0.131* |
|
|
|
|
|
Grade |
0.004 |
|
|
|
|
|
Only child |
-0.004 |
|
|
|
|
|
Anxious parenting style |
-0.009 |
|
|
|
|
|
Refusal parenting style |
0.016*** |
|
|
|
|
|
Over-protection |
0.005 |
|
|
|
|
|
Negative affect experience |
0.043*** |
|
|
|
|
|
Negative coping style |
0.008 |
|
|
|
* p < 0.05, ** p < 0.01, *** p < 0.001.
Finally, multiple hierarchical regression analysis was performed with non-physical bullying total score as the dependent variable. In the first step, the control variable is introduced into the regression equation to construct model 1; in the second step, the negative parenting style is also introduced into the regression equation to construct model 2; in the third step, negative affect experience and negative coping style are introduced into the equation, and model 3 is constructed. Model 1 indicated that 2.1% of the variance of the perceived non-physical bullying could be attributed to sex and only child (F = 3.448, p < 0.05). Among them, sex has a significant negative predictive effect on non-physical bullying total score (β=-0.146, p<0.05), and only child has a significant positive predictive effect on non-physical bullying total score (β=0.148, p<0.05). Model 2 indicated that 9.6% of the variance of the perceived non-physical bullying could be attributed to refusal parenting style of the negative parenting style (F = 10.719, p < 0.001). Refusal parenting style has a significant positive predictive effect on non-physical bullying total score (β=0.016, p<0.001). Model 3 indicated that 11.3% of the variance of the perceived non-physical bullying could be attributed to negative affect experience (F = 20.643, p < 0.001). Negative affect experience can significantly and positively predict non-physical bullying total score (β=0.043, p<0.001).
3.4. Mediation Analysis
Figure 2. The chain-like mediating effect of negative affect experience and negative coping styles between refusal parenting style and the perceived verbal bullying.
Based on the literature research and the results of multiple hierarchical regression analysis in Table 4-1, a structural equation model of the relationship between refuasl parenting style, negative affective experience, negative coping style and perceived verbal bullying was constructed. Figure 2 presents the results of a test of the influence on the perceived verbal bullying in the upper-grade primary school students, with the refusal parenting style as an independent variable and negative affect experience and negative coping styles as mediating variables. Table 5 shows the model goodness-of-fit indices. Among them, CMIN/DF = 2.052, RMSEA = 0.046, CFI = 0.903, and GFI = 0.922 were all within their acceptable ranges, which indicated that the model fit well and that negative affect experience and negative coping styles presented a chain-like mediating effect between the refusal parenting style and the perceived verbal bullying in the upper-grade primary school students.
Table 5. The model fitting goodness index table with the perceived verbal bullying as the dependent variable.
|
Goodness-of-Fit Indices |
CMIN/DF |
CFI |
IFI |
GFI |
AGFI |
RMSEA |
|
Result |
2.052 |
0.903 |
0.904 |
0.922 |
0.905 |
0.046 |
Based on the literature research and the results of multiple hierarchical regression analysis in Table 4-1, a structural equation model of the relationship between refuasl parenting style, negative affective experience and perceived cyberbullying was constructed. A mediation effect test model was shown ( in Figure 3), with the perceived cyberbullying as the dependent variable ,the refusal parenting style as an independent variable and the negative affect experience as a mediating variable. Table 6 shows the model goodness-of-fit indices. Among them, CMIN/DF = 2.515, RMSEA = 0.056, CFI = 0.934, and GFI = 0.940, which were all within their acceptable ranges; this indicated that the model fit well. It was shown that negative affect experience had a partial mediating effect between the refusal parenting style and the perceived cyberbullying in the upper-grade elementary school students.
Figure 3. Partial mediating effect of negative affect experience between the refusal parenting style and the perceived cyberbullying.
Table 6. The model fitting goodness index table with the perceived cyber bullying as the dependent variable.
|
Goodness-of-Fit Indices |
CMIN/DF |
CFI |
IFI |
GFI |
AGFI |
RMSEA |
|
Result |
2.515 |
0.934 |
0.935 |
0.919 |
0.940 |
0.056 |
Based on the literature research and the results of multiple hierarchical regression analysis in Table 4-2, a structural equation model of the relationship between refuasl parenting style, negative affective experience and perceived relationship bullying was constructed. A mediation effect test model was shown ( in Figure 4) , with the perceived relationship bullying as the dependent variable , the refusal parenting style as an independent variable, negative affect experience as a mediating variable, Table 7 shows the model goodness-of-fit indices. Among them, CMIN/DF = 2.523, RMSEA = 0.056, CFI = 0.936, and GFI = 0.940, which were all within their acceptable ranges; this indicated that the model fit well. It was shown that negative affect experience had a partial mediating effect between the refusal parenting style and the perceived relationship bullying in the upper-grade elementary school students.
Figure 4. Partial mediating effect of negative affect experience between the refusal parenting style and the perceived relationship bullying.
Table 7. The model fitting goodness index table with the perceived relationship bullying as the dependent variable.
|
Goodness-of-Fit Indices |
CMIN/DF |
CFI |
IFI |
GFI |
AGFI |
RMSEA |
|
Result |
2.523 |
0.936 |
0.936 |
0.919 |
0.940 |
0.056 |
Based on the literature research and the results of multiple hierarchical regression analysis in Table 4-2, a structural equation model of the relationship between refuasl parenting style, negative affective experience and perceived non-physical bullying total score was constructed. A mediation effect test model was shown ( in Figure 5), with the perceived non-physical bullying total score as the dependent variable, the refusal parenting style as the independent variables and the negative affect experience as the mediating variable. Table 8 shows the model goodness-of-fit indices. Among them, CMIN/DF = 2.132, RMSEA = 0.048, CFI = 0.916, and GFI = 0.922, which were all within the acceptable range; this indicated that the model fit well. It was shown that negative affect experience had a partial mediating effect between the refusal parenting style and the perceived school non-physical bullying in the upper-grade elementary school students.
Figure 5. Partial mediating effect of negative affect experience between the refusal parenting style and the perceived non-physical bullying on school.
Table 8. The model fitting goodness index table with the perceived non-physical bullying total score as the dependent variable.
|
Goodness-of-Fit indices |
CMIN/DF |
CFI |
IFI |
GFI |
AGFI |
RMSEA |
|
Result |
2.132 |
0.916 |
0.917 |
0.904 |
0.922 |
0.048 |
Discussion
Point 1: The discussion requires an opening paragraph that describes the study.
Point 2: The discussion should be organized by the hypotheses, not by the analyses performed.
Point 3: The discussion should integrate the results with past published research and provides a clear explanation of the results, including their importance.
Response 1,2,3: In this study, we aimed to explore the impact of parenting styles on the students’ perceived school non-physical bullying, and the effects of negative affect experience and negative coping style between them. Correlation analysis shows that negative affect experience, negative coping style, negative parenting style and the perceived school non-physical bullying are all positively related with each other, These findings are consistent with those of recent studies [61, 72, 73]. Wójcik & Rzeńca(2021)found Significant link between negative coping and being bullied[61]. Other research reported that bullying victim recognition was significantly correlated with the degree of parent-child relationship (Zhao et al., 2021) [ 72]. Rauschenberg et al. (2020) found bullied people reported extremely strong negative emotional experiences [73]. These findings confirmed hypotheses 1 ,2 and 4.
The results of multiple hierarchical regression analysis showed that, after controlling the confounding variables such as sex, grade, and only child or not, refusa1 parenting style, negative emotional experience and negative coping style still had significant influence on students' perceived school non-physical bullying.
The results of mediating effect analysis by structural equation modeling showed that negative affect experiences and negative coping styles had a chain-like mediating effect between the refusal parenting style and students’ perceived school verbal bullying. Th findings confirmed hypotheses 6. Moreover, negative affect experience had a partial mediating effect between the refusal parenting style and students’ perceived school non-physical bullying total score , relationship bullying and cyberbullying [72, 74]. These findings partially confirmed hypothesis 3,7and 8. In this study only hypothesis 5 has not been confirmed.
Point 4: The discussion does not include: generalizability of the findings, limitations and future research.
Response 4:
- Limitations and Future Research
Based on the theoretical basis of previous research, this paper studies and analyzes the relationship between the parenting style of fifth- and sixth-grade students and the perceived school non-physical bullying by means of interviews and questionnaires, and discusses the role of negative affect experience and negative coping style in the above relationship. The topic selection has certain research innovation, but due to the limitations of practice and theory, there are still some deficiencies that need to be further improved by follow-up research.
First of all, this research mainly adopts the questionnaire survey method, which may be interfered by some unclear factors, so the results of this research may be affected to a certain extent. Subsequent research can use more abundant and rigorous research methods to explore, For example, pairing the bully and the bullied to explore the reasons for the differences in the behaviors emitted and felt by two different subjects in the same behavior, thereby improve the accuracy of research results.
Secondly, from the perspective of the research object, the object of this study is the fifth and sixth grade students of 8 primary schools in Zhejiang Province. Although the sample is representative to a certain extent, the generality of the research results needs to be further tested in the follow-up research. Future research can expand the scope of sample selection and select sample groups from different regions and cities to improve the representativeness of research sampling and further verify the results of this study.
Thirdly, refusal” type of parenting might also impact children’s self-concept and make them act in ways in which others perceive them differently and make them more likely to be a target of bullying. This is also the direction that future research needs to further verify.
References
Point 1: Authors should include more recent references.
Response 1:
- Guo, L., Wang, W., Li, W. et al. Childhood maltreatment predicts subsequent anxiety symptoms among Chinese adolescents: the role of the tendency of coping styles. Transl Psychiatry. 2021,11(1), 340. https://doi.org/10.1038/s41398-021-01463-y.
- Zhao, Y., Hong, J. S., Zhao, Y., & Yang, D. Parent–child, teacher–student, and classmate relationships and bullying victimization among adolescents in china: implications for school mental health. School Mental Health, 2021, 13(3), 1-11.
- Rauschenberg, C., Os, J. V., Goedhart, M., Schieveld, J., & Reininghaus, U. Bullying victimization and stress sensitivity in help-seeking youth: findings from an experience sampling study. European Child & Adolescent Psychiatry, 2020, 1-15.
- Lee, J. Pathways from childhood bullying victimization to young adult depressive and anxiety symptoms. Child Psychiatry & Human Development, 2021, 52(4).
- Meldrum, R. C., Patchin, J. W., Young, J. T. N., & Hinduja, S. Bullying victimization, negative emotions, and digital self-harm: testing a theoretical model of indirect effects. Deviant Behavior, 2020, 43(3), 303-321.
- M Wójcik, & Krzysztof Rzeńca. Disclosing or hiding bullying victimization: a grounded theory study from former victims' point of view. School Mental Health, 2021, 13(4),808-818.
- Choi, B., & Park, S. Bullying perpetration, victimization, and low self-esteem: examining their relationship over time. Journal of Youth and Adolescence, 2021, 50(4), 739-752.
- Li, R., Yao, M., Liu, H., & Chen, Y. Relationships among autonomy support, psychological control, coping, and loneliness: comparing victims with nonvictims. Personality and Individual Differences, 2019, 138, 266-272.
Thank you very much for your guidance and help!